# Regulation of the macrolide resistance ABC-F translation factor MsrD

Corentin R. Fostier [1], Farès Ousalem[1], Elodie C. Leroy[2], Saravuth Ngo[1], Heddy Soufari[2,3], C. Axel Innis [2], Yaser Hashem [2] ✉ & Grégory Boël [1] ✉

Antibiotic resistance ABC-Fs (ARE ABC-Fs) are translation factors that provide resistance against clinically important ribosome-targeting antibiotics which are proliferating among pathogens. Here, we combine genetic and structural approaches to determine the regulation of *streptococcal* ARE ABC-F gene *msrD* in response to macrolide exposure. We show that binding of cladinose-containing macrolides to the ribosome prompts insertion of the leader peptide MsrDL into a crevice of the ribosomal exit tunnel, which is conserved throughout bacteria and eukaryotes. This leads to a local rearrangement of the 23 S rRNA that prevents peptide bond formation and accommodation of release factors. The stalled ribosome obstructs the formation of a Rho-independent terminator structure that prevents *msrD* transcriptional attenuation. Erythromycin induction of *msrD* expression via MsrDL, is suppressed by ectopic expression of *mrsD*, but not by mutants which do not provide antibiotic resistance, showing correlation between MsrD function in antibiotic resistance and its action on this stalled complex.

ABC-F ATPases are members of the ATP-Binding Cassette (ABC) superfamily. Unlike other ABC proteins, the ABC-Fs are translation factors and some of them, termed antibiotic resistance ABC-Fs (ARE ABC-Fs), confer resistance to clinically important antibiotics that target the ribosomal peptidyl transferase center (PTC) and/or the nascent peptide exit tunnel (NPET)[1–7]. ABC-F proteins are composed of two ABC domains joined by a linker domain called P-site tRNA-interaction motif (PtIM)[8,9], also termed antibiotic resistance determinant (ARD) for ARE ABC-Fs[1–3]. The two ABC domains dimerize upon binding of two ATP molecules and in this conformation the factor can bind to the ribosomal E site[9], where the PtIM adopts an α-helical hairpin conformation that directly interacts with the peptidyl-tRNA and extends toward the PTC/NPET. Three antibiotic resistance phenotypes are associated with ARE ABC-Fs: (i) Macrolides, Ketolides, Streptogramins group B (MKS$_B$); (ii) Pleuromutilins, Lincosamides, Streptogramins group A (PLS$_A$); (iii) Phenicols, Oxazolidinones (PhO)[10,11]. However, despite structural investigations[1–3,5,7], some questions remain about the exact molecular mechanism of action of ARE ABC-Fs.

Over the last 40 years, biochemical and structural investigations demonstrated that the nascent chains (NC,) which elongate into the ribosomal tunnel, can interact with metabolites or antibiotics, thus adapting protein synthesis to environmental cues[12–16]. In human pathogens, antibiotic-dependent formation of stalled ribosome complexes (SRCs) on regulatory ORFs, named leader peptides, can subsequently allow the regulation of downstream resistance genes in response to antibiotic exposure[17–20].

The ARE ABC-F gene *msrD* is a member of the *msr* (macrolide and streptogramin B resistant) gene group previously referred to as *mel*[21,22]. This gene is generally found in operon with a macrolide efflux facilitator (*mefA* or *mefE*) gene and the two corresponding proteins act synergistically to confer macrolide resistance[23–26]. This operon is a part of the Macrolide Efflux Genetic Assembly (MEGA)[27,28] that disseminates in human pathogens on numerous mobile genetic elements: conjugative transposons (tn916-type)[29] and conjugative prophages (Φ1207.3)[30] among *Streptococci*; conjugative plasmids (pMUR050)[31] among *Enterobacteria*. In *Streptococci*, the operon is transcribed in the

---

[1]Expression Génétique Microbienne, CNRS, Université Paris Cité, Institut de Biologie Physico-Chimique, 75005 Paris, France. [2]ARNA Laboratory, UMR 5320, U1212, Institut Européen de Chimie et Biologie, Univ. Bordeaux, Centre National de la Recherche Scientifique, Institut National de la Santé et de la Recherche Médicale, 33607 Pessac, France. [3]Present address: SPT Labtech Ltd., SG8 6HB Melbourn, United Kingdom. ✉e-mail: yaser.hashem@inserm.fr; boel@ibpc.fr

presence of erythromycin (ERY) as a polycistronic mRNA from a single promoter located ~350 bp upstream of the *mefA*[32,33] gene (Fig. 1a). It is also regulated by ribosome-mediated transcriptional attenuation via the MefAL leader peptide (MTASMRLR), which is closely related to the ErmDL leader peptide (MTHSMRLRFPTLNQ), both polypeptides harboring the characteristic macrolide-arrest motif Arg/Lys-X-Arg/Lys, so called "+x +" motif[14,33–35].

Here, we report the presence of a second transcriptional attenuator on the *mefA/msrD* operon regulating exclusively the *msrD* expression upon macrolide exposure. The presence of ERY induces ribosomal stalling on the previously identified *msrDL* leader peptide (encoding MYLIFM)[34], allowing RNA polymerase (RNAP) to bypass an intrinsic terminator. Our findings demonstrate that the stalling occurs due to hampered translation termination on the UAA stop codon, and inhibition of action of both release factors 1 and 2 (RF1 and RF2). Our results are supported by the cryo-electron microscopy (cryo-EM) structure of MsrDL-SRC that provides molecular insights on how the PTC precludes productive accommodation of RF1/RF2 and how the NC discriminates tunnel-bound cladinose-containing macrolides. The path of the NC within the tunnel greatly differs from previously described leader peptides[12–16] with the NC latching into a ribosomal crevice, that was shown to be conserved from prokaryotes to eukaryotes. This crevice is delimited by the 23 S rRNA nucleotides U2584, U2586, G2608 and C2610 that may form a ribosomal functional site. Finally, our results demonstrate that the two ATPase sites of MsrD are functionally asymmetric and the protein can negatively self-regulate its own synthesis in the presence of ERY.

## Results

### MsrD provides macrolide and ketolide resistance

The *mefA/msrD* macrolide resistance operon (Fig. 1a) is part of the MEGA element currently spreading among clinical and livestock isolates[33,36–38]. A phylogenetical analysis (Supplementary Fig. 1a) shows

that the operon disseminates predominantly in Gram-positive firmicutes (mostly *Streptococci*) and in some Gram-negative proteobacteria (such as *Haemophilus influenzae*, *Neisseria gonorrhoeae* or *Escherichia coli*). The gene *msrD* shares ~62 % identity with its closest homolog *msrE*[7], both factors exhibiting the canonical ABC-F domains organization (Fig. 1b and Supplementary Fig. 1b).

The model bacterium *Escherichia coli* (*E. coli*) presents undeniable advantages for genetic and molecular biology, but the presence of multidrug efflux systems greatly limits its use in antibiotic research[39,40]. To circumvent this limitation, we took advantage of the *E. coli* DB10 strain, which exhibits exacerbated sensitivity to macrolide antibiotics[41,42]. Heterologous expression of the wild-type (WT) gene *msrD* from an arabinose-inducible promoter conferred macrolide and ketolide resistance phenotype to the strain (Table 1), demonstrating the functionality of the factor in this organism. Expression of $msrD_{WT}$ resulted in eightfold minimal inhibitory concentration (MIC) increase for the 14-membered macrolide ERY, 8-fold for the 15-membered macrolide azithromycin (AZI) and 2-fold for telithromycin (TEL, a ketolide antibiotic derived from the 14-membered macrolides). Similar results were reported for *Streptococcus pneumoniae* clinical isolates[23]. However, no change in MIC was observed for the 16-membered macrolides tylosin (TYL) and spiramycin (SPI), or the non-macrolide antibiotics lincomycin (LNC), linezolid (LNZ) and retapamulin (RTP) (Table 1).

To understand the function of the ATPase activity of MsrD, we constructed ATPase-deficient variants $MsrD_{E125Q}$, $MsrD_{E434Q}$ and $MsrD_{E125Q/E434Q}$ (hereafter referred as $MsrD_{EQ2}$) by replacing the catalytic glutamic acid residue (E) in the Walker B motif of each or both ABC domains by a glutamine (Q) (Fig. 1b and Supplementary Fig. 1b). This mutation strongly reduces the ATP hydrolysis while preserving the local stereochemistry of the active site and has been extensively used to trap ABC-F factors on the ribosome in ATP-bound conformation[1–3,5,9]. Protein expression levels have been assessed by western blot for each tested variants (Supplementary Fig. 1c).

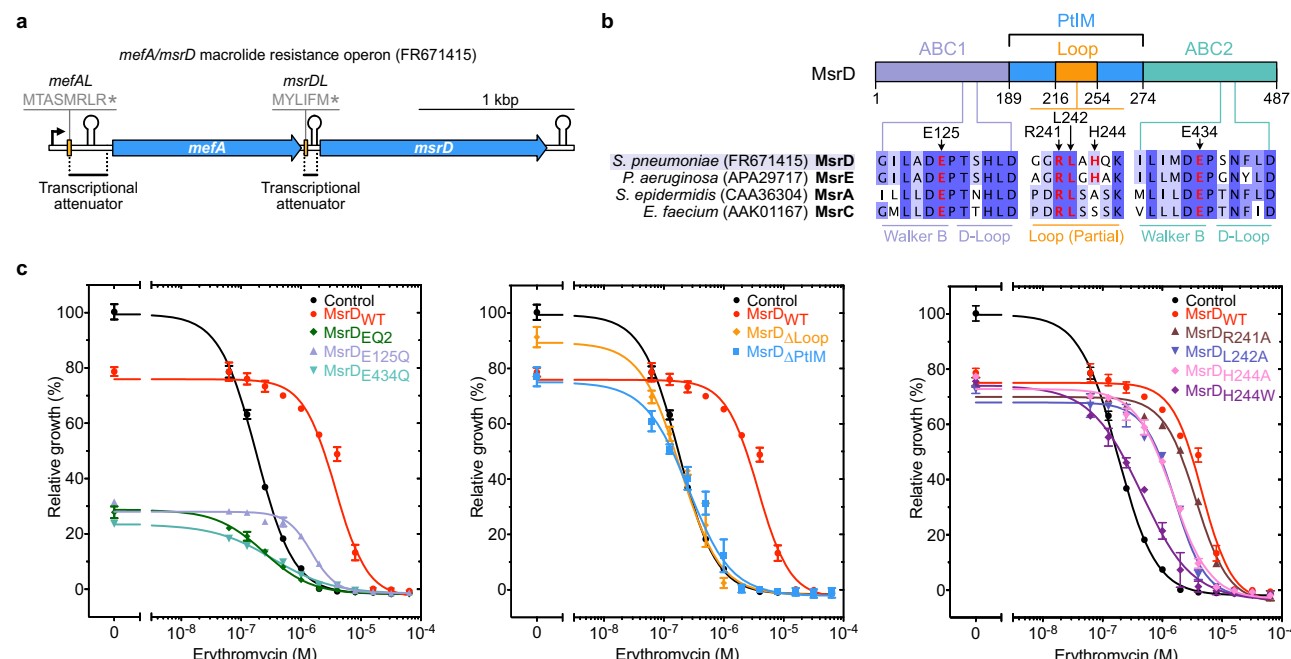

**Fig. 1 | Translation factor MsrD alleviates erythromycin effects upon translation in vivo. a** Organization of *mefA/msrD* macrolide resistance operon. In the presence of ERY, ribosomes stall during translation of *mefAL* leading to transcription anti-attenuation, both *mefA* and *msrD* being then transcribed. Similar mechanism occurs during translation of *msrDL*, which regulates transcription of *msrD* only. **b** Sequence alignment of various *msr* homologs visualized with Jalview according to percentage of identity. Positions of tested mutations are highlighted

in red and indicated by arrows. Main features of ABC-F proteins are indicated on the schematic. Genebank accession numbers are indicated between brackets. See also Supplementary Fig. 1b. **c** Relative growth of *E. coli* DB10 expressing *msrD* variants in the presence of ERY after 24 h. Optical densities were normalized relative to optical densities of *E. coli* DB10 pBAD-*Control* grown in the absence of ERY. Error bars represent mean ± s.d. for triplicate experiments. See also Supplementary Table 1. Source data are provided as a Source Data file.

**Table 1 | Minimum inhibitory concentration (MIC) and half maximal inhibitory concentration (IC$_{50}$) of *E. coli* DB10 expressing *msrD*$_{WT}$ in the presence of ribosome-targeting antibiotics**

| Antibiotic | pBAD-*Control* | | pBAD-*msrD*$_{WT}$ | |
|---|---|---|---|---|
| | MIC (μM) | IC$_{50}$ (μM) | MIC (μM) | IC$_{50}$ (μM) |
| Erythromycin | 2 | 0,179 ± 0,007 | 16 | 4,592 ± 0,582 |
| Azithromycin | 0.25 | 0,046 ± 0,002 | 2 | 0,953 ± 0,053 |
| Telithromycin | 1 | 0,086 ± 0,008 | 2 | 0,714 ± 0,066 |
| Tylosin | 4 | 1,649 ± 0,151 | 4 | 1,672 ± 0,103 |
| Spiramycin | 2 | 0,758 ± 0,063 | 2 | 0,81 ± 0,078 |
| Lincomycin | 32 | 12,61 ± 2,154 | 32 | 12,62 ± 1,27 |
| Linezolid | 32 | 13,62 ± 2,064 | 32 | 13,41 ± 1,137 |
| Retapamulin | 1 | 0,071 ± 0,004 | 1 | 0,075 ± 0,007 |

See Methods for experimental details. Source data are provided as a Source Data file.

We evaluated the influence of E-to-Q mutations of MsrD on bacterial growth and antibiotic resistance phenotypes and showed that all the mutations impair both phenotypes (Fig. 1c). While MsrD$_{E125Q}$ and MsrD$_{EQ2}$ mutations had a similar effect on bacterial growth in the absence or at low ERY concentration, the MsrD$_{E434Q}$ mutant was more toxic. However, MsrD$_{E125Q}$ maintained some resistance (MIC = 4 μM) compared to control (MIC = 2 μM). This observation was supported by the half maximal inhibitory concentration (IC$_{50}$) measurement of the MsrD$_{E125Q}$ strain which was ~8 times higher than Control (Fig. 1c and Supplementary Table 1). The two ATP hydrolysis sites (Supplementary Fig. 1b) were therefore concluded to perform different tasks and are functionally asymmetric, as previously reported for the ARE ABC-F Vga(A)[43].

We also evaluated the importance of other MsrD residues. The truncation of the PtIM (replaced by three glycines) or just the Loop (ΔPtIM, ΔLoop respectively) completely abolished the resistance phenotype in vivo (Fig. 1c and Supplementary Fig. 1e). This has been previously reported in vitro for the MsrE[7]. Punctual mutations in the Loop (Fig. 1c) affected the resistance phenotype to a different extent. Mutation of residue R241 to alanine had almost no effect on the phenotype, but equivalent mutation for residues L242 or H244 reduced the IC$_{50}$ by ~3 fold (Fig. 1c and Supplementary Table 1). This demonstrated that these residues are important, but are not essential for MsrD function despite being predicted to be located at the vicinity of the antibiotic (Supplementary Fig. 1d) in the MsrE-ribosome complex[7]. Interestingly, this finding contrasts with the mutagenesis assays for MsrE, showing that the mutation R241A reduced the resistance by more than 50%[7]. Replacement of residue H244 by the larger residue tryptophan should displace the drug by steric occlusion, but the variant lost most of its antibiotic resistance phenotype (Fig. 1c). Similarly, MsrD$_{WT}$ does not provide resistance to TYL nor SPI (Table 1), despite predicted clash of both antibiotics with MsrD according to the MsrE structure (Supplementary Fig. 1d). The same observation was made with LNC, LNZ and RTP (Table 1 and Supplementary Fig. 1d). Therefore, MsrD's action on the antibiotic is indirect, and direct steric occlusion to displace the antibiotic seems unlikely.

**Ribosome-mediated transcriptional attenuation regulates *msrD* expression**

Analysis of the *mefA/msrD* operon sequence indicates a stringent ribosome binding site (RBS) only 8 bp downstream of the gene *mefA* stop codon. This sequence was proposed to be required for the translation of the downstream putative small ORF, *msrDL* (Supplementary Fig. 2a). In order to test the influence of *msrDL* on *msrD* expression, a fluorescent reporter with the sequence spanning from the first nucleotide downstream *mefA* stop codon to *msrD* first three codons fused to the Yellow Fluorescent Protein gene (YFP) was cloned in the low copy pMMB

plasmid[44] under the control of an IPTG-inducible P$_{LlacO-1}$ promoter[45]. The resulting plasmid, pMMB-*msrDL*-*msrD*$_{(1-3)}$:*yfp*, was used to directly follow the fluorescence signal reflecting the *yfp* expression (Fig. 2a and Supplementary Fig. 2b).

A basal fluorescence level was found to be stable in the *E. coli* DB10 strain containing pMMB-*Control* independently of ERY concentration (Fig. 2a), while a fluorescence increase that correlated with the increase in ERY concentration was observed in cells containing the plasmid pMMB-*msrDL*-*msrD*$_{(1-3)}$:*yfp*. This indicated an ERY-dependent induction of *msrD*. Optimal induction was found at 100 nM ERY, however, 6.25 nM was sufficient to significantly induce *msrD*$_{(1-3)}$:*yfp* expression, demonstrating high sensitivity of the system toward its inducer. A reduced, but significant fluorescence signal was detected at 0 nM ERY indicating an imperfect repression of the regulation. Inactivation of the ORF *msrDL* by replacing the start codon with an amber stop codon (pMMB-*msrDL*$_{(no\ ORF)}$-*msrD*$_{(1-3)}$:*yfp*) resulted in the elimination of the fluorescence induction (Fig. 2a). This demonstrated that *msrDL* translation is an important and necessary *cis*-acting feature that regulates *msrD*$_{(1-3)}$:*yfp* expression as previously reported for *ermC*[46].

In bacteria, antibiotic-dependent gene induction relies mainly on translational or transcriptional attenuation[47]. To determine if *msrDL* prompts a transcriptional or translational attenuation, northern blot analysis with a probe located in *msrD*$_{(1-3)}$:*yfp* 3' UTR was performed on total RNA extracted from bacteria with or without ERY and/or IPTG exposure. A 1.2-kb band corresponding to the *msrDL*-*msrD*$_{(1-3)}$:*yfp* mRNA was detected in the presence of IPTG and started to increase incrementally 5 min after exposure to 100 nM ERY (Fig. 2b). Consistent with this ERY-dependent transcript accumulation, we concluded that *msrD* regulation by *msrDL* occurs at the transcription level.

Transcription attenuation employs premature transcription termination via: (i) Rho-dependent terminator (RDT) by binding of Rho factor, a RecA-type ATPase, to a C-rich and G-poor sequence known as Rho utilization site[48]; (ii) Rho-independent terminator (RIT) where a stable GC-rich stem-loop followed by a poly-uridine sequence causes RNAP to drop off[49]. Treatment with bicyclomycin, a Rho inhibitor, did not result in constitutive expression of the reporter, thus excluding the possibility of a RDT attenuation (Supplementary Fig. 2c). Search for secondary structures in the intergenic region between the genes *msrDL* and *msrD* using ARNold server[50] revealed the presence of a RIT with a ΔG of −7.9 kcal.mol$^{-1}$. We also identified a consensus "TTNTTT" NusG-dependent RNAP pausing site embedded in the *msrDL* sequence (Fig. 2c and Supplementary Fig. 2a), as previously described for *vmlR*[51,52]. In order to test possible transcription termination activity, we generated a *msrDL*-*msrD* construct that lacks the putative terminator (referred as pMMB-*msrDL*$_{(no\ term)}$-*msrD*$_{(1-3)}$:*yfp*). The deletion of the terminator resulted in a constitutive expression of the fluorescent reporter in the absence of ERY, indicating that the terminator is a necessary *cis*-acting regulatory feature that terminates transcription (Fig. 2c). These results validate the presence of a second transcriptional attenuator on the *mefA/msrD* operon that regulates exclusively the gene *msrD*.

**Selective drug-sensing by MsrDL prevents its translation elongation and termination**

To determine how this regulation occurs via a ribosome-mediated mechanism, translation of *msrDL* mRNA with the PURE system[53] was subjected to various antibiotics and its propensity to form SRC was analyzed by toeprinting[54] (Fig. 3 and Supplementary Fig. 3). A strong toeprint signal was observed in the presence of ERY and AZI, while it was absent in the presence of TEL, TYL and SPI (Fig. 3a). This in vitro result correlates to the in vivo observations made with the *msrD*$_{(1-3)}$:*yfp* reporter gene which was also only induced by ERY and AZI, but not by TEL, TYL, SPI and non-macrolide antibiotics (Fig. 3b).

We used an inhibitor of the first peptide bond formation, which allows the identification of an ORF's start codon[55], to further

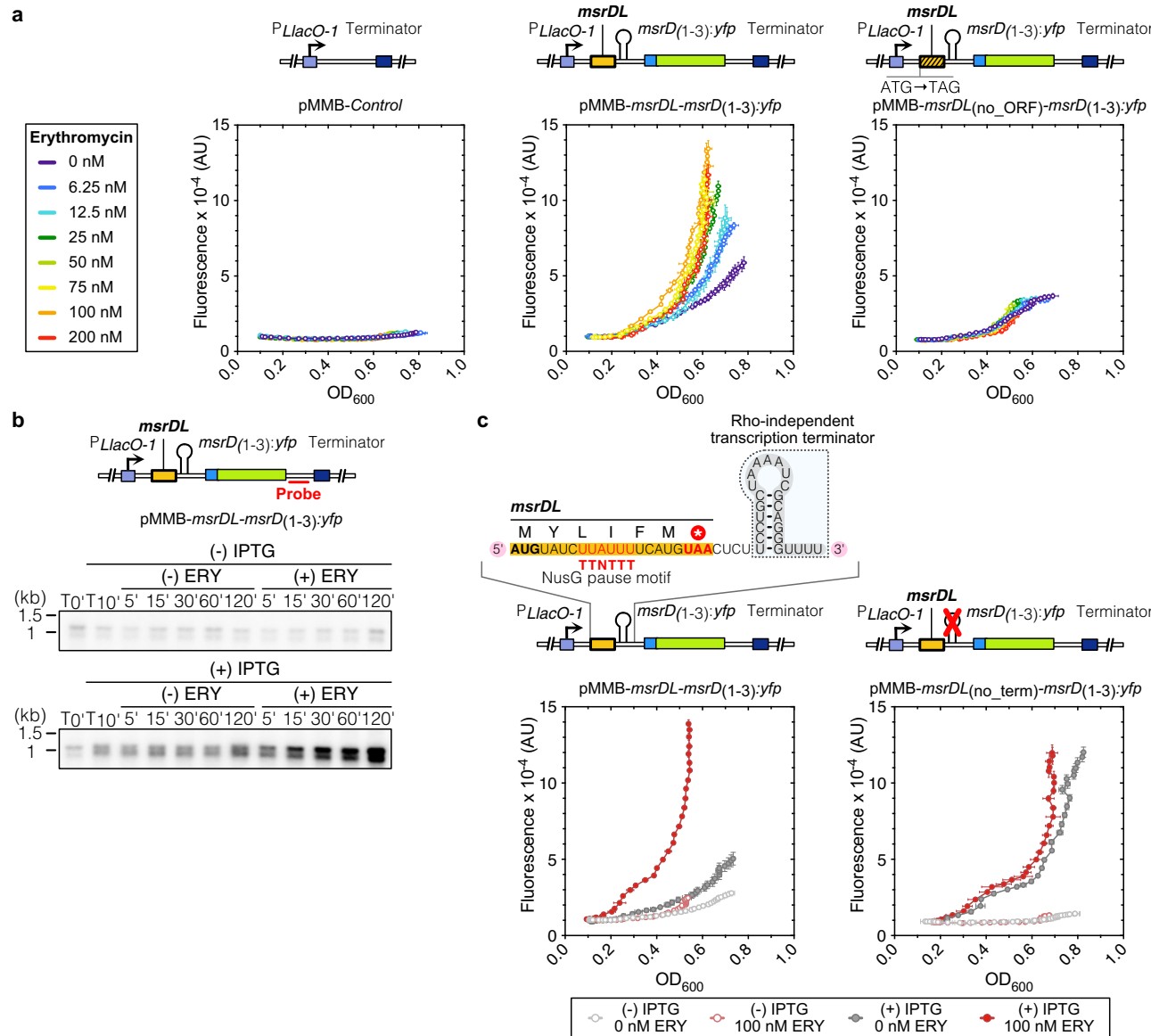

**Fig. 2 | Erythromycin-dependent transcriptional attenuation regulates *msrD* expression. a** ERY-dependent induction of *msrD*(1-3):*yfp*. Fluorescent reporters shown on schematics have been introduced in *E. coli* DB10 and grown in presence of 1 mM IPTG and increasing sublethal ERY concentrations during 17 h. Fluorescence has been plotted against OD$_{600}$, error bars for both axes represent mean ± s.d. for triplicate experiments. **b** Northern analysis of *msrD* transcript. RNAs were extracted before adding or not 1 mM IPTG (T$_{0'}$), 10 min after adding IPTG (T$_{10'}$), and 5, 15, 30, 60, 120 min after adding or not 100 nM ERY. Location of probe in *msrD*(1-3):*yfp* 3' UTR is shown on the schematic. The presence of a second band was also observed, but we hypothesized that it was an abortive transcript resulting from the construct insofar as its presence correlated with induction by IPTG and ERY. Note the presence of leaky transcription in absence of IPTG, that is slightly amplified in presence of ERY. **c** Deletion of the intrinsic terminator between *msrDL* and *msrD*(1-3):*yfp* lead to a constitutive induction in absence of ERY. Error bars for both axes represent mean ± s.d. for triplicate experiments. Source data are provided as a Source Data file.

characterize the translation of *msrDL* mRNA. We showed that in presence of 50 μM RTP, a clear toeprint can be observed (Fig. 3c). This toeprint corresponds to a stalled ribosome displaying the initiation codon in its ribosomal P site, which confirms that these ribosomes can initiate *msrDL* translation. The toeprint observed at positions +31, +32 and +33 relative to *msrDL* 5' end (Fig. 3a, c) in presence of ERY and AZI indicates SRC formation located at the termination step of MsrDL synthesis with the C-terminal methionine (M6) codon in the P site and an UAA stop codon in the A site. In accordance with this, reactions that are depleted of release factors (RF1, RF2, RF3) (Fig. 3c) showed the same toeprint results. Moreover, the addition of puromycin led to the loss of this toeprint signal, while preserving it in the presence of ERY (Fig. 3c). Puromycin is an aminonucleoside antibiotic mimicking an A site substrate tyrosyl-tRNA and causing premature chain termination.

While actively translating ribosomes are sensitive to this antibiotic, stalled ones tend to be refractory[15,16,20,56,57]. The resistance of MsrDL-SRC to puromycin is consistent with an ERY-sensing NC that leads to PTC silencing and stalling. In consequence, the stalled ribosome on *msrDL* covers the first half of the hairpin of the RIT and therefore prevents its formation and transcription termination (Fig. 3c and Supplementary Fig. 3c).

The main dissimilarity between tested macrolides resides in the presence or absence of a cladinose moiety on C3 of the macrocyclic lactone ring which appeared to be critical for MsrDL-SRC formation (Fig. 3a) similarly to ErmCL[20]. Absence of significant toeprint in the presence of TEL (whose C3 cladinose sugar is replaced by a keto group) suggested that *msrDL* is translated efficiently without stalling. TYL and SPI differ from the other macrolides by the presence of disaccharide

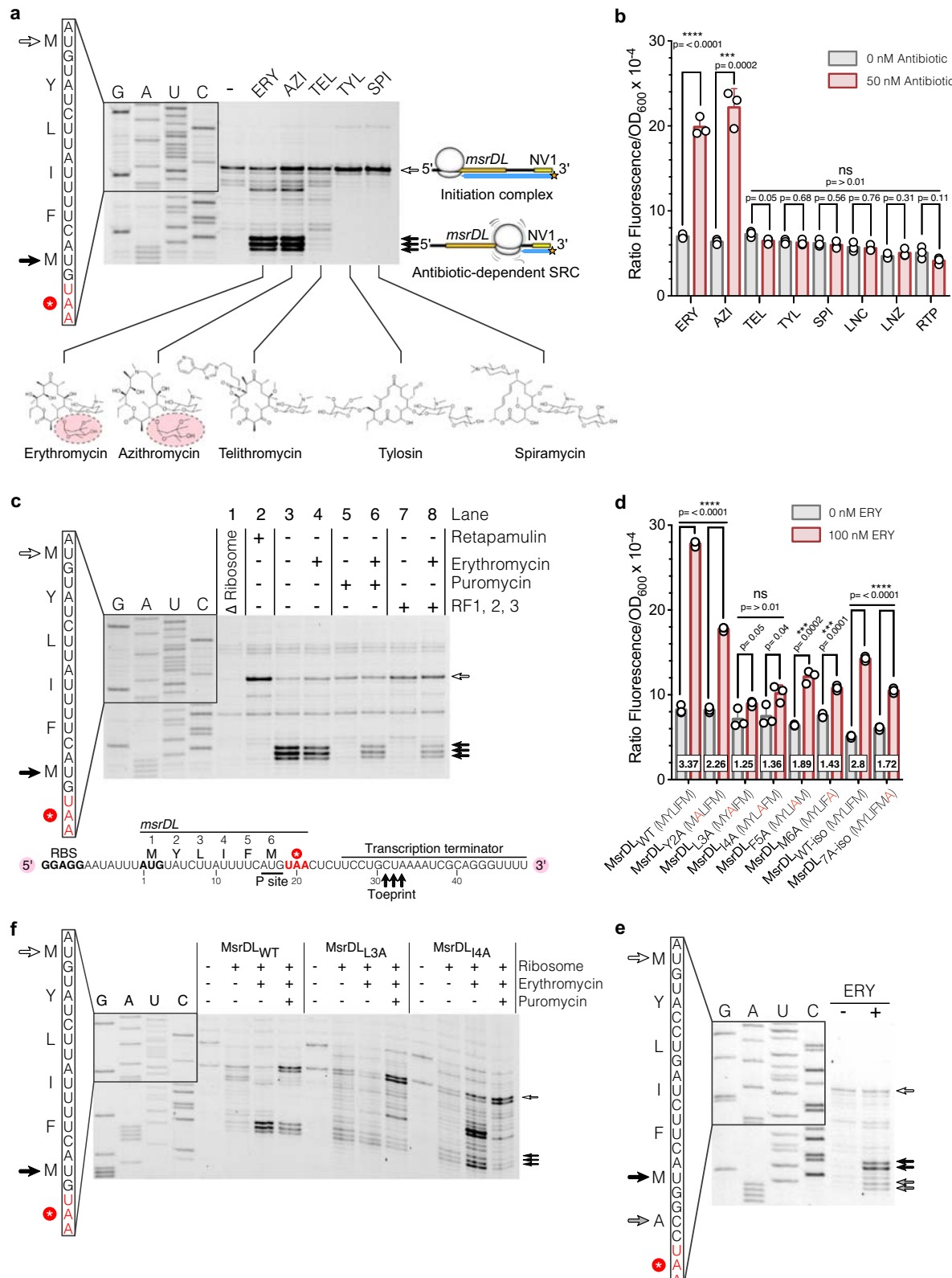

mycaminose-mycarose in place of the C5 monosaccharide deso-samine. When bound to the ribosome, this extended sugar moiety protrudes into the PTC of the ribosome[58]. Absence of intermediate toeprint and increase of the toeprint corresponding to initiation complex in the presence of these two drugs imply that these anti-biotics stabilize the initiation complex. Previous results[58–60] have shown that they inhibit the formation of the peptide bond, our results

suggest that it occurs mainly for the first peptide bond forma-tion (Fig. 3a).

We replace individually each amino acid of MsrDL sequence by an alanine to investigate the importance of each residue and its ability to induce the *msrD*(1-3):*yfp* reporter, which indirectly reports the forma-tion of an ERY-dependent SRC (Fig. 3d). The scan showed a strong reduction of inducibility for all the mutated residues with the

**Fig. 3 | MsrDL is a macrolide-sensing nascent chain that stalls the ribosome.**
**a** Toeprinting assay of *msrDL* in the absence (-) or in the presence of 50 μM of various macrolide antibiotics. White arrow indicates initiation codon. Black arrows indicate ribosome stalling, with M6 codon in the P site. Chemical structure of antibiotics is shown, C3 cladinose sugar of ERY and AZI being highlighted. See also Supplementary Fig. 3. **b** In vivo induction of *msrD*(1:3):*yfp* by various PTC/NPET targeting antibiotics. Bacteria containing pMMB-*msrDL*-*msrD*(1:3).*yfp* were grown for 17 h in the presence of 1 mM IPTG, in the absence (gray histograms) or in the presence of 50 nM antibiotics (red histograms). **c** Toeprinting assay depleted of release factors was performed in the absence of ribosome (line 1), in the presence of 50 μM RTP to assess start codon (line 2), without or with 50 μM ERY (line 3 and 4), without or with 50 μM ERY then supplemented with 100 μM puromycin (line 5 and 6), without or with 50 μM ERY in the presence of RF1/RF2/RF3 (line 7 and 8). The schematic indicates position of toeprint signal on the synthetic mRNA, P site codon of MsrDL-SRC is underlined. See

also Supplementary Fig. 3. **d** Effects of *msrDL* variants on the expression of *msrD*(1:3).*yfp*. Bacteria were grown during 17 h in the presence of 1 mM IPTG, in the absence (gray histograms) or in the presence of 100 nM ERY (red histograms). Square boxes show fold of induction. **e** Toeprinting assay performed on the MsrDL_WT, MsrDL_L3A and MsrDL_I4A constructs in the absence or presence of 50 μM ERY or 100 μM puromycin. Open arrow indicates the initiation codon. Black arrows indicate ribosomal stalling, with the M6 codon in the P site. **f** Toeprinting assay performed on the MsrDL_7A-iso construct in the absence or presence of ERY. Addition of a sense codon after M6 codon lead to translational arrest with M6 codon in the P site (black arrows) and A7 codon in the A site in the presence of ERY. A faint toeprint was also observed with A7 codon in the P site (gray arrows) and the stop codon in the A site. **b**, **c** Error bars represent mean ± s.d. for triplicate experiments and the *p* values were determined by unpaired two-sided *t*-test without adjustment. Source data are provided as a Source Data file.

exception of residue Y2 (Fig. 3d). Since MsrDL is a short hexapeptide, it is very likely that most of the residues would have certain importance in maintain the conformation that is able to sense ERY. A complete loss of inducibility was observed when the residues L3 and I4 were replaced by alanine, suggesting that these two residues are directly involved in drug-sensing. Toeprinting assays conducted on these two constructs showed that both have less MsrDL-SRC formation in the presence of ERY and the stalled complexes were not resistant to puromycin (Fig. 3e). The importance of these two residues were structurally confirmed and is detailed in the next section.

Considering that MsrDL inhibits the action of both RF1 and RF2 in the presence of ERY, we mutated UAA stop codon into RF1-specific UAG and RF2-specific UGA stop codons to investigate a putative preferential inhibition as previously shown for TnaC[61]. Mutation of UAA into UAG or UGA reduced the level of inducibility of the reporter gene (Supplementary Fig. 2d) demonstrating the importance of the stop codon. This observation was corroborated by our phylogenetic analysis, which revealed a large prevalence of UAA stop codon among the different *msrDL* variants (Supplementary fig. 2a). Most likely the UAA stop codon has been selected by evolution to maximize *msrD* expression in the presence of an inducer. We designed a construct of MsrDL_WT reporter where the codons of *msrDL* were replaced by synonymous codons, as described in the Methods section. This construct, MsrDL_WT-iso, showed a reduction of the *yfp* expression, possibly due to the lack of NusG pausing site, but the regulation is mostly preserved (Fig. 3d). Therefore, the amino acid composition of the peptide is responsible for the translational stalling. Addition of an extra alanine before the stop codon in the MsrDL_7A-iso construct reduced the regulatory potential of the leader peptide (Fig. 3d). Toeprinting assay revealed that this construct has two stalling positions: one occurs with the additional A7 codon in the ribosomal A site and the second with the stop codon in the A site (Fig. 3f). Thus, MsrDL peptide seems to induce efficient ribosomal stalling by preventing elongation and termination. The MsrDL_7A-iso construct showed less induction of the *yfp* reporter by ERY compared to MsrDL_WT-iso construct (Fig. 3d) suggesting that when inhibition of elongation and termination does not occur at the same location, the regulation is less efficient.

## Molecular mechanism of ribosome stalling by MsrDL

To understand how MsrDL inhibits translation termination in the presence of ERY, a synthetic mRNA containing a single *msrDL* copy was translated in vitro with purified *E. coli* DB10 ribosomes in the presence of ERY (as described in Supplementary Fig. 4a) and the resulting sample was subjected to cryo-EM. After image processing and particle images sorting two nearly identical subclasses were obtained and refined, one containing only a P-site tRNAs and one containing P- and E-site tRNAs (11% and 28.1% of total particles, respectively), as described in Methods and in Supplementary Fig. 4b. However, the ribosome (observed in the non-rotated state) and the MsrDL peptide conformations in both classes are identical, therefore

they were merged together in order to improve the final resolution. The resulting initial map of the 70 S ribosome (EMD-13805) with an average resolution of 3.0 Å that was subjected to multibody refinement, generating reconstructions for the body and the head of the small subunit (EMD-13807 and EMD-13808) and the large subunit (EMD-13806), presenting average resolutions of 3.08, 3.3 and 2.97 Å, respectively (Supplementary Fig. 4b, 4c and Supplementary Table 2). The resolutions of the derived reconstructions were consistent with unambiguous assignment of ribosomal proteins side chains, rRNA nucleotides and some post-transcriptional modifications (Supplementary Fig. 4d-4f). The local resolution of the codon-anticodon allowed unambiguous identification of the P-site tRNA as an elongator _MettRNA^Met (Supplementary Fig. 4e), which was also confirmed by the presence of distinctive elements such as N⁴-acetylcytidine at position 34 (ac⁴C34) and N⁶-threonylcarbamoyladenosine at position 37 (t⁶A37) (Supplementary Fig. 4e, f)[62,63], consistent with the toeprint results which assigned the M6 codon in the P site and the UAA stop codon in the A site (Fig. 3c).

A clear density corresponding to an ERY molecule bound in its canonical position was found at the entrance of the NPET (Fig. 4a, b and Supplementary Fig. 4d), allowing attribution of macrocyclic lactone ring, cladinose and desosamine sugars. A continuous density at the 3' end of the P-site tRNA extending within the entrance of the ribosome tunnel was identified as MsrDL-NC and the local resolution of ~3 Å allowed modeling of the peptide de novo (Fig. 4b and Supplementary Fig. 4d). The leader peptide adopted a hook-like shape with its N-terminal extremity protruding into a cavity at the entrance of the tunnel delimited by 23 S rRNA nucleotides U2584, U2586, G2608 and C2610 (Fig. 4b). One noticeable feature of MsrDL interaction with the ribosome is its residue Y2 that forces the base U2584 to bulge out and Y2 forms a π-stacking with the base G2583 (Fig. 4c, d) in place of U2584. This unique interaction should stabilize the peptide conformation, while it appears structurally significant, it is not strictly essential, since replacement of the Y2 by an alanine reduces the induction of the reporter gene by less than 50% (Fig. 3d). In addition, this residue is substituted in some MsrDL variants (Supplementary Fig. 2a). MsrDL did not show numerous significant electrostatic interactions with the ribosome, since it is mostly composed of hydrophobic amino acids. It is likely stabilized by hydrophobic interactions.

Our biochemical investigations showed that the MsrDL-SRC was formed due to hampered translation termination on the UAA stop codon. This stop codon is recognized by both RF1 and RF2, which subsequently catalyze peptide hydrolysis via their conserved GGQ motif protruding within the PTC. Structural alignment of our atomic model to RF1- and RF2-containing ribosome structures[64] revealed that proper accommodation of the RFs is prevented by steric clashes of the methylated glutamine and the glycines of the GGQ motif against the residue F5 of MsrDL and 23 S rRNA bases U2584 and U2585, respectively (Fig. 4e). The conformation of MsrDL peptide stabilized the PTC in an uninduced conformation[65] which prevents the opening of the

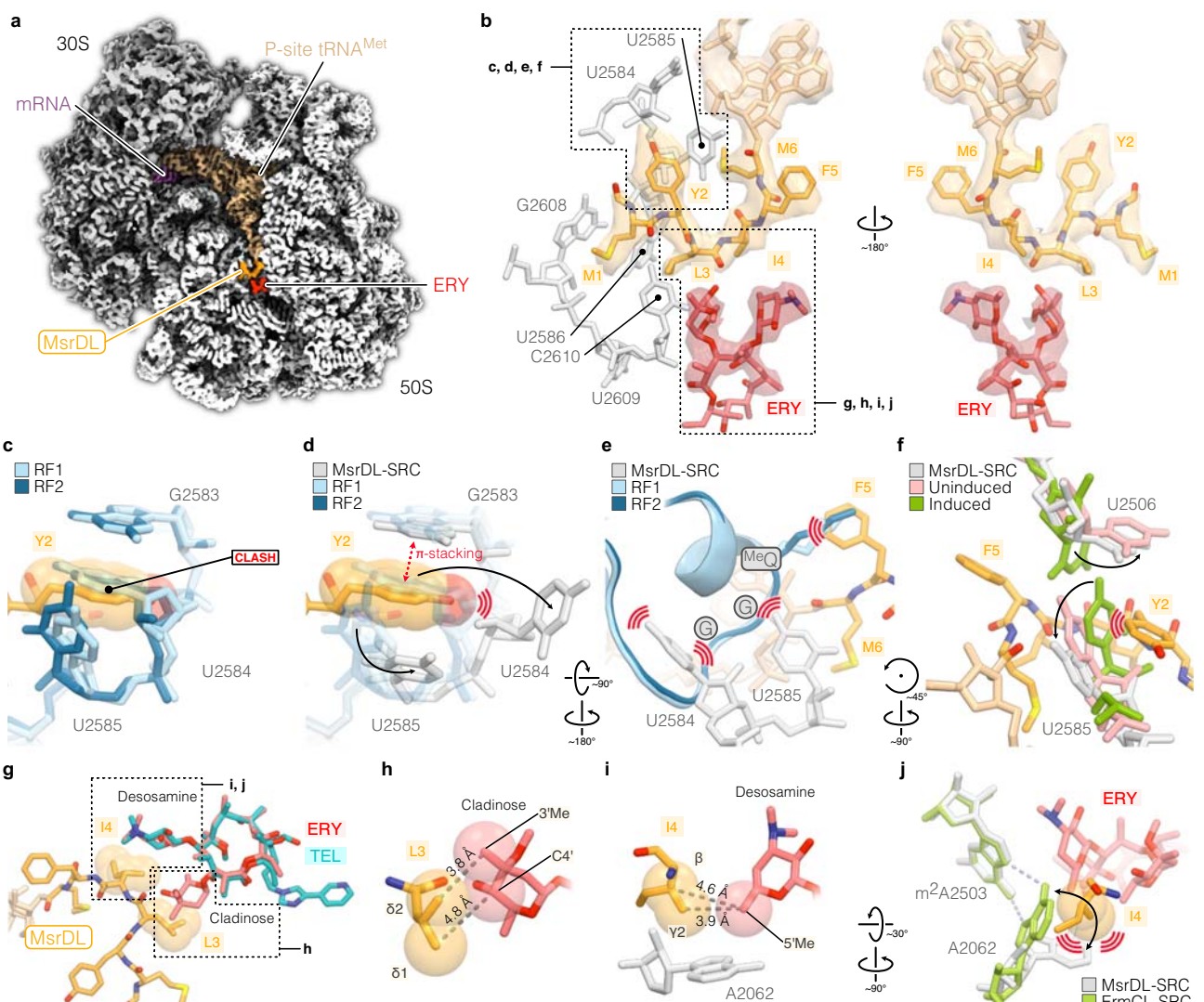

**Fig. 4 | Structure of MsrDL-SRC. a** Transverse section of the cryo-EM map showing the 30 S (gray) and 50 S (white) ribosomal subunits, mRNA (purple), ERY (red) and MsrDL-NC (gold) bound to P-site tRNA^Met (beige). **b** Close-up of MsrDL-NC within ribosomal tunnel showing experimental density and modeled structure, colored as Fig. 4a while 23 S rRNA nucleotides are shown in light gray. **c, d** Presence of MsrDL residue Y2 displace nucleotide U2584 out of its position when compared to RF1- (PDB 5J30, light blue) and RF2-containing (PDB 5CZP, blue) termination complex, thus forming a π-stacking with G2583[64]. **e** Conformation of U2584 and U2585 prevented productive accommodation of RF1 (PDB 5J30, light blue) and RF2 (PDB 5CZP, blue) while catalytic methylated glutamine clashed with MsrDL residue F5[64].

**f** PTC in MsrDL-SRC (light gray) is stabilized in an uninduced state (PDB 1VQ6, pink) rather than in an induced state (PDB 1VQN, green) as U2585 is pushed back by MsrDL residue Y2[65]. **g–i** Molecular basis for C3 cladinose sugar recognition by MsrDL. Residue L3 at proximity of cladinose sugar while residue I4 at proximity of desosamine sugar. TEL lacking cladinose sugar and failing to form MsrDL-SRC has been aligned (PDB 4V7S)[76]. **j** Presence of residue I4 avoided rotation of A2062 to form an Hoogsteen base pairing with m²A2503 as is the case for ErmCL-SRC (PDB 3J7Z, green)[12]. Light blue dashed lines indicate hydrogen bonds formed by Hoogsteen base pairing. For the whole figure, structures were aligned on domain V of 23 S rRNA. Spheres represent van der Waals radii.

active site necessary for peptide bond formation and hydrolysis (Fig. 4f). This observation suggests that the MsrDL-SRC cannot catalyze peptide bond formation, as confirmed by the toeprinting performed on the MsrDL_{7A-iso} construct (Fig. 3e). Structural alignment of the MsrDL-SRC with ErmBL- and ErmCL-SRCs demonstrated that 23 S rRNA nucleotides U2506 and U2585 are not in the same conformation (Supplementary Fig. 5a, b). These bases in MsrDL-SRC adapt a conformation similar to ErmDL-, SpeFL- and TnaC(R23F)-SRCs (Supplementary Fig. 5c–e)[12–16]. However, the unusual position of base U2584 seems to be a unique feature of MsrDL-SRC.

Our in vivo and in vitro experiments demonstrated that the MsrDL monitors the presence of cladinose-containing drugs. Residues L3 and I4 being critical for drug recognition and/or conformation of the peptide (Fig. 3d, e). Accordingly, in our atomic model we identified residue L3 as the closest amino acid to the cladinose moiety and

residue I4 as the closest to the desosamine moiety of the ERY molecule (Fig. 4b, g). However, similarly to ErmBL, no close contact between the nascent chain and the drug was observed[13,66]. The residue L3 being ~3.8 Å away from cladinose sugar and the residue I4 being ~3.9 Å away from desosamine sugar, consistent with a hydrophobic interaction between MsrDL and the drug (Fig. 4g–i). This observation is also in accordance with previous descriptions of pentapeptides conferring ERY resistance in which the presence of a leucine or isoleucine at position 3 was critical for drug recognition[67,68]. Moreover, the importance of 23 S rRNA residue A2062 acting as an ERY-sensor and contributing in silencing the PTC, has been demonstrated for some regulatory leader peptides[69,70]. In the case of ErmCL, the critical feature for macrolide-dependent stalling is the rotation of the base A2062 that forms an Hoogsteen base pairing with m²A2503[12,69,70]. The structure of the MsrDL-SRC showed that residue I4 restrains rotation of base A2062

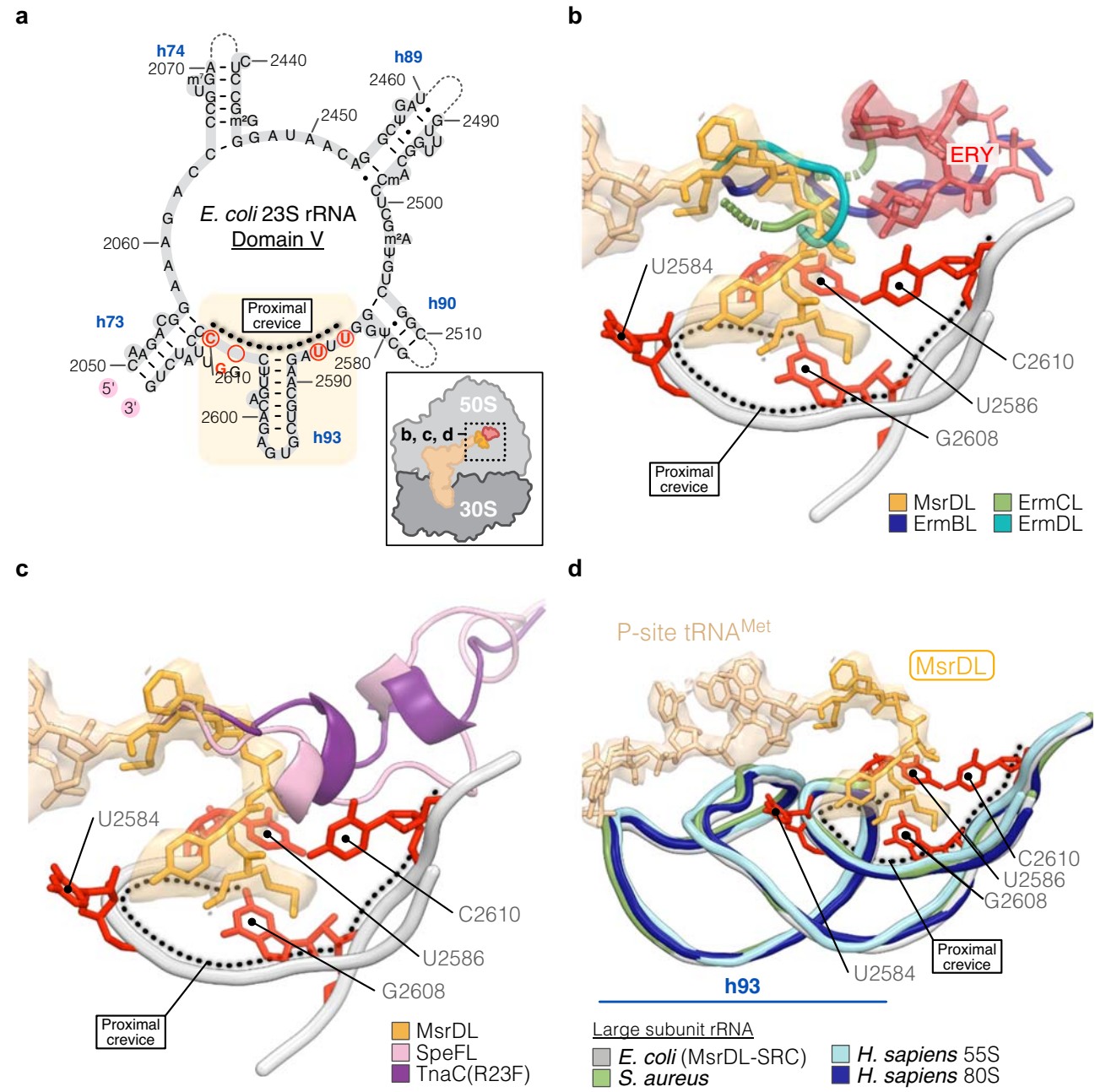

**Fig. 5 | MsrDL engages in a conserved crevice at the NPET entrance. a** Secondary structure of the *E. coli* 23 S rRNA domain V showing location of the proximal crevice at the base of h93. For the whole figure, nucleotides delimiting proximal crevice are shown in red. **b, c** Comparison of MsrDL path (gold) with ERY-dependent leader peptides ErmBL (PDB 5JTE, blue), ErmCL (PDB 3J7Z, green), ErmDL (PDB 7NSO, teal) as well as L-ornithine-sensing SpeFL (PDB 6TC3, pink) and L-tryptophan-sensing TnaC(R23F) (PDB 7O1A, purple)[12–16]. See also Supplementary Fig. 5a–e. **d** Proximal crevice in domain V is conserved from bacteria to human (MsrDL-SRC, gray; *S. aureus* PDB 6YEF, light green; *H. sapiens* 55 S mitoribosome PDB 7A5F, light blue; *H. sapiens* 80 S ribosome PDB 6OLI, marine blue)[93–95]. See also Supplementary Fig. 5g and h. For the whole figure, atomic model of MsrDL is shown with its experimental density. Structures were aligned on domain V of 23 S rRNA.

(Fig. 4j). Therefore, MsrDL induces an A2062/m²A2503-independent ribosome stalling[70].

The path of MsrDL-NC within the ribosomal tunnel contrasts with all previous descriptions of elongating polypeptides (Figs. 4b and 5). Indeed, ERY-sensing ErmBL, ErmCL and ErmDL leader peptides[12–14] engage in the NPET and bypass the tunnel-bound drug, while MsrDL curves before encountering it and then engages into a dead-end crevice at the entrance of the NPET (Figs. 4b and 5b). The same observation (Fig. 5c) can be made with metabolite-sensing SpeFL/TnaC(R23F) leader peptides[15,16], which avoid the crevice and enter the NPET. The first methionine of MsrDL-NC latches into the dead-end crevice, which is delimited by 23 S rRNA nucleotides U2584, U2586,

G2608 and C2610 (Fig. 5a–c). This crevice located at the base of helix 93 (h93) is part of domain V of 23 S rRNA and is conserved from bacteria to human (mitoribosome and cytosolic ribosome) (Fig. 5d, Supplementary Fig. 5f and 5g). We termed it "proximal crevice".

## MsrD negatively regulates its own expression

The protein MsrD provides antibiotic resistance by interacting with antibiotic-stalled ribosome while *msrDL* translation in the presence of ERY leads to the formation of a SRC. If MsrD is also able to rescue this SRC, it will repress its own expression and form a feedback loop similar to the ARE ABC-F Vga(A)[71]. We tested this hypothesis by using an *E. coli* DB10 strain double transformed with pMMB-*msrDL-msrD*(1-3):*yfp* and

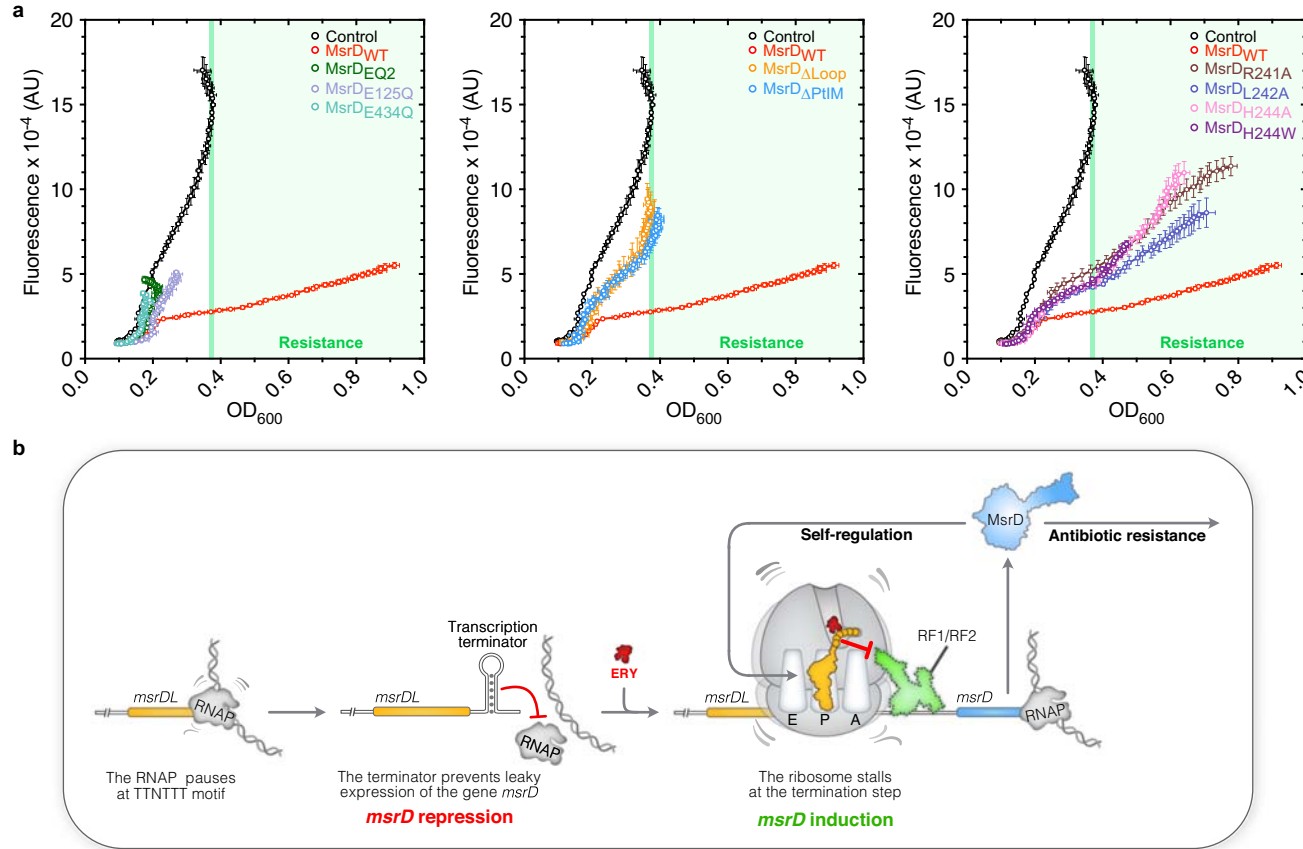

**Fig. 6 | MsrD negatively regulates its own synthesis upon erythromycin exposure. a** Effects of MsrD variants on MsrDL. *E. coli* DB10 containing pMMB-*msrDL*-*msrD*$_{(1-3)}$:*yfp* and expressing various *msrD* mutants were grown in the presence of 0.2 % L-Arabinose, 1 mM IPTG and 300 nM ERY, both OD$_{600}$ and fluorescence being recorded over 24 h. Fluorescence was plotted against OD$_{600}$, error bars for both axes represent mean ± s.d. for triplicate experiments. Color code is same as Fig. 1c. Light green rectangle indicates bacterial growth over control plasmid. **b** Model of MsrD regulating its own expression and providing antibiotic resistance. The

presence of a NusG-dependent pause site might stall the RNAP and explains why the system works in bacteria where transcription and translation are not so tightly coupled. In the absence of ERY, RNAP dropped off at Rho-independent transcription terminator. In the presence of ERY, the ribosome following the RNAP (presumably paused) stalled and unwound the terminator leading to *msrD* transcription. Once translated, MsrD negatively regulates its own expression on one side, and provides antibiotic resistance on the other side. Source data are provided as a Source Data file.

the compatible pBAD plasmid expressing various *msrD* variants. Strains were grown in the presence of 300 nM of ERY with their optical density and fluorescence monitored (Fig. 6a and Supplementary Fig. 6). The strain carrying the pBAD-*Control* showed a high fluorescence and a low OD$_{600}$ as observed in Fig. 2a, reflecting an induction of the reporter and a susceptibility to ERY. On the contrary, expression of *msrD*$_{WT}$ led to a higher cell growth and a stronger reduction of fluorescence, which correspond respectively to antibiotic resistance and a repression of the expression of the reporter. These results support a model where MsrD$_{WT}$ can rescue MsrDL-SRC and therefore alleviate the RIT-mediated repression on the *msrD* transcription.

The strain expressing *msrD*$_{E125Q}$ had a low ERY resistance (Fig. 1c). When it was co-transformed with the pMMB-*msrDL*-*msrD*$_{(1-3)}$:*yfp* plasmid, no resistance was observed, possibly because the presence of the two plasmids created some toxicity. Overall, the results showed that MsrD variants which lost the ability to provide antibiotic resistance (ΔPtIM, ΔLoop and ATPase deficient mutants) also lost the ability to rescue MsrDL-SRC (Fig. 6a and Supplementary Fig. 6). Point mutations in the Loop of the PtIM indicated some discrepancy between antibiotic resistance phenotype and MsrDL-SRC rescue. All the mutants showed an intermediary phenotype with less antibiotic resistance and less repression of the reporter compared to MsrD. The mutant MsrD$_{H244w}$ had the same repression effect on the reporter as MsrD$_{H244A}$ for an equivalent OD$_{600}$, but it provided more than 3 times less antibiotic resistance (Figs. 1c and 6a). The mutant MsrD$_{R241A}$ provides a

resistance close to MsrD, but does not rescue the MsrDL-SRC as efficiently as the latter one (Figs. 1c and 6a). Mutants MsrD$_{L242A}$ and MsrD$_{H244A}$ provided similar resistance, but MsrD$_{H244A}$ conferred a stronger repression of the reporter (Figs. 1c and 6a). Together these observations demonstrate that the determinants of the dual function of MsrD in conferring antibiotic resistance and rescuing MsrDL-SRC are overall similar.

## Discussion

We used genetic, molecular biology and structural approaches to dissect the regulation of the streptococcal ARE ABC-F gene *msrD*. We characterized the *msrDL* regulatory ORF which induces ribosome stalling in the presence of cladinose-containing macrolides (Figs. 3, 4 and 6b). Our structural data shows that this inhibition results from the conjoint effects of: (*i*) MsrDL maintaining 23 S rRNA bases U2506 and U2585 in an uninduced conformation (Fig. 4f), (*ii*) the steric occlusion by U2584, U2585 and MsrDL residue F5 that precludes the productive accommodation of RF1 and RF2 (Figs. 3c and 4e). The MsrDL-stalled ribosome then hampers the formation of a RIT and thus allows for the transcription of *msrD* (Figs. 2c, 3a, 3c and 6b). The coupling between translation and transcription necessary for this mechanism in *Streptococci* is possibly mediated by a NusG-dependent RNAP pausing site in *msrDL* (Fig. 2c).

Alongside ErmBL, ErmCL and ErmDL, MsrDL constitutes the fourth ERY-sensing NC that has been structurally described, however

some mechanisms of drug-sensing and PTC silencing differ. Firstly, MsrDL and ErmCL need the presence of the C3 cladinose sugar of ERY for ribosomal stalling (Fig. 3a, b), this differs to ErmBL and ErmDL which induce stalling with drugs lacking this sugar such as oleandomycin, solithromycin or telithromycin[12,14,66,72]. Secondly, similar to ErmBL, no close contact between the drug and MsrDL was observed (Fig. 4h, i)[66]. Thirdly, MsrDL is the only described leader peptide that exploits inhibition of its hydrolysis by RFs to induce stalling in the presence of antibiotic, comparable to the metabolite sensors TnaC and SpeFL[15,16]. ErmBL[13], ErmCL[12] and ErmDL[14] employ only elongation inhibition in physiological condition. Finally, MsrDL-NC conformation and path within the ribosome differ from previously described ligand-sensing leader peptides that elongate within the tunnel (Fig. 5b, c)[12–16]. It adopts a hook-like shape possibly due to hydrophobic interactions with the drug (Fig. 4b, g–i), while its initiating methionine engages in a dead-end crevice at the NPET entrance (Fig. 5). This crevice named "proximal crevice" is conserved in prokaryotes and eukaryotes (Supplementary Fig. 5f and 5g) and is a functional site within the ribosome.

ARE ABC-F genes tend to be transcriptionally regulated as illustrated by $msrD$, $vmlR$, $vgaA$, $lmoO919$ and $lmrC$[51,71,73–76]. However, MsrD can regulate its own expression by creating a negative feedback-loop which is not the case for VmlR[51]. The gene $msrD$ forms an operon with the efflux pump $mefA$ (Fig. 1a), which is also regulated via ribosome-mediated transcriptional attenuation, translational stalling occurring on the $mefAL$[33]. Therefore, $msrD$ is under the dual-regulation of two ERY-sensing leader peptides. One explanation for this redundancy may reside in the need for the bacteria to tightly regulate $msrD$ because of its toxicity. The expression of $msrD_{WT}$ in $E.\ coli$ DB10 led to a 20% growth defect compared to control condition (Fig. 1c), suggesting that $msrD_{WT}$ expression has a fitness cost for the bacteria and can be beneficial only in the presence of antibiotic. The presence of two attenuators creates a double-lock system to avoid any basal expression in the absence of inducer and the negative feedback-loop maintains a minimal amount of MsrD production. Thus, as long as MefA can export enough antibiotic to provide resistance, $msrD$ should be kept repressed. The fitness cost of the expression of ARE ABC-F genes is a potential Achilles heel of the bacteria containing those genes and may be exploited to fight antibiotic resistance. MsrD provides resistance to ERY, AZY and TEL[23] (Table 1), but only the first two induce its expression, indicating that MsrD can provide some resistance for an antibiotic which does not induce its expression. Thus, MsrDL facilitates the use of MsrD by the bacteria and maintains its evolution under its control.

## Methods

### Construction of plasmids
Strains and plasmids used in this study are listed in Supplementary Table 3. Oligonucleotides are listed in Supplementary Table 4. For the whole study, the considered reference sequence for $mefA/msrD$ macrolide resistance operon was a Tn916-type transposon inserted in $Streptococcus\ pneumoniae$ strain 23771 genome (Genbank accession number FR671415.1)[77]. Plasmids pBAD-$Control$ and pBAD-$msrD_{WT}$ containing a C-terminal hexahistidine tag (originally referred as pBAD33 and pVN50) were retrieved from Olivier Chesneau at Institut Pasteur (Paris, France)[24]. Both $mefA/msrD$ macrolide resistance operon and pBAD-$msrD_{WT}$ encode the same protein MsrD (Uniprot accession number A0A496M710). pBAD plasmids allow a tight and stringent expression control via catabolite repression, inserted gene being repressed in presence of 0.4% (w/v) Glucose and induced in the presence of 0.2% (w/v) L-Arabinose. Transformed clones were selected with 20 µg.ml$^{-1}$ chloramphenicol. In this study, all $msrD$ mutants contain a C-terminal hexahistidine tag. Catalytic mutant $msrD_{EQ2}$ was generated by amplifying pBAD-$msrD_{WT}$ with primer pairs 5/6 and 7/8, both fragments being assembled with NEBuilder HiFi DNA Assembly Master Mix (New England Biolabs). Catalytic mutants $msrD_{E125Q}$ and $msrD_{E434Q}$ were generated via quickchange mutagenesis with primers

pairs 9/10 and 11/12. Mutants $msrD_{\Delta Loop}$ and $msrD_{\Delta PtlM}$ were generated based on phylogenetic alignments by deleting residues between K216/K254 and E189/A274 respectively, a three-glycine linker being added to allow flexibility, via quickchange mutagenesis using primer pairs 13/14 and 15/16. Variants $msrD_{R241A}$, $msrD_{L242A}$, $msrD_{H244A}$ and $msrD_{H244W}$ were generated via fusion PCR by amplifying $msrD_{WT}$ with primers pairs 1/18, 1/20, 1/22, 1/24 and 2/17, 2/19, 2/21, 2/23 respectively. Backbone was amplified with primer pair 3/4 and fragments were assembled with NEBuilder HiFi DNA Assembly Master Mix (New England Biolabs). All pMMB constructs originated from a low copy IPTG-inducible plasmid pMMB67EH-$yfp$, consisting of the original pMMB67EH[44] containing optimized $venus$-$yfp$ under the control of P$_{tac}$ promoter (manuscript in preparation). Transformed clones were selected with 100 µg.ml$^{-1}$ ampicillin. To test $msrD$ functions in vivo, several fluorescent reporter genes have been designed. First, pMMB67EH-$yfp$ native P$_{tac}$ promoter was replaced by P$_{LlacO-1}$ promoter as previously described[45]:

5'- ATAAATGTGAGCGGATAACATTGACATTGTGAGCGGATAACAA-GATACTGAGCAC**A** −3' (lac operators, shaded gray; transcription start, bold). This IPTG-inducible promoter allows a tight and stringent transcription regulation compared to P$_{tac}$ promoter, insofar as no regulatory element is found in the 5′ UTR. Promoter was replaced by amplifying pMMB67EH-$yfp$ with primer pairs 25/26, PCR fragment being then re-circularized via NEBuilder HiFi DNA Assembly Master Mix (New England Biolabs). The resulting plasmid was named pMMBpLlacO-1-67EH-$yfp$. Plasmid pMMB-$msrDL$-$msrD_{(1–3)}$:$yfp$ has been designed by introducing the sequence spanning from the first nucleotide downstream $mefA$ stop codon to $msrD$ three first codons fused to $yfp$. The introduced sequence is as follows: 5′-ACAATATT**GGAGGA**ATATTT*ATGTATCTTATTTTCATG TAA*CTCTTCCTGCTAAAATCGCAGGGTTTTCCCTGCATACAAGCAAATG AAAGCATGCGATTATAGACA**GGAGGA**AATGTT<u>ATGGAATTA</u>−3′ (RBS, bold; $msrDL$, italic; $msrD$ three first codons are underlined). To clone such construct, $yfp$ gene was amplified from pMMB67EH-$yfp$ with primer pairs 27/29 then with 28/29 to generate the insert, backbone was amplified with primer pair 30/31 using pMMBpLlacO-1-67EH-$yfp$ as matrix, both fragments being then assembled with NEBuilder HiFi DNA Assembly Master Mix (New England Biolabs). Plasmid pMMB-$control$ was built by removing the $msrDL$-$msrD_{(1–3)}$:$yfp$ cassette via quickchange mutagenesis using primer pair 32/33, the resulting construction containing only the P$_{LlacO-1}$ promoter followed by the plasmid endogenous $rrnB$ transcription terminator. Plasmid pMMB-$msrDL_{(no\_term)}$-$msrD_{(1-3)}$:$yfp$ (where the RIT between $msrDL$ and $msrD$ is deleted) was generated by amplifying pMMB-$msrDL$-$msrD_{(1–3)}$:$yfp$ with primer pair 34/35, PCR fragment being then re-circularized via NEBuilder HiFi DNA Assembly Master Mix (New England Biolabs). The various $msrDL$ mutants were cloned via quickchange mutagenesis by amplifying pMMB-$msrDL$-$msrD_{(1–3)}$:$yfp$ with primers 36 to 51. Plasmid pMMB-$msrDL_{(MYLIFMA-isocodons)}$-$msrD_{(1–3)}$:$yfp$ was generated by replacing $msrDL_{WT}$ sequence (ATGTATCTTATTTTCATGTAA) by a recoded sequence (ATGTACCTGATCTTCATGGCCTAA) using isocodons and introducing an extra alanine codon (7 A codon) before stop codon. Sequence was recoded using isocodons because mutating the WT sequence introduced a new promoter. To do so, pMMB-$msrDL$-$msrD_{(1–3)}$:$yfp$ was amplified with primer pair 52/53, PCR fragment being then re-circularized via NEBuilder HiFi DNA Assembly Master Mix (New England Biolabs). Recoded sequence without the 7 A codon (ATG-TACCTGATCTTCATGTAA) was generated via quickchange mutagenesis by amplifying pMMB-$msrDL_{(MYLIFMA-isocodons)}$-$msrD_{(1–3)}$:$yfp$ with primer pair 54/55, leading to plasmid pMMB-$msrDL_{(WT-isocodons)}$-$msrD_{(1–3)}$:$yfp$.

### Antibiotic susceptibility testing, MIC and IC50 determination
A saturated preculture of $E.\ coli$ DB10 transformed with pBAD plasmid was grown overnight at 37 °C under vigorous shaking in Luria-Bertani Miller broth (LB), 20 µg.ml$^{-1}$ chloramphenicol and supplemented with 0.4% (w/v) Glucose. Antibiotic susceptibility testing assay was

performed in Mueller-Hinton broth (MH, Sigma Aldrich), antibiotics being diluted via serial dilutions. A 96-wells flat-bottom plate (Flacon) was filled with a final volume per well of 200 µl, containing 20 µg.ml⁻¹ chloramphenicol and 0.2% (w/v) L-Arabinose and antibiotic to test. Wells were inoculated at $OD_{600}$ ~ 0.03-0.04 prior to addition of 60 µl mineral oil (Sigma Aldrich) avoiding evaporation but not oxygen diffusion. Plates were therefore incubated for 24 h in CLARIOstar Plus plate reader (BMG Labtech) at 37 °C with 600 rpm double-orbital shaking, $OD_{600}$ being measured each 30 min. Optical densities at 24 h were then normalized relative to optical densities of *E. coli* DB10 pBAD-*Control* grown in the absence of antibiotic in Prism 7(GraphPad). MIC was determined as absence of growth compared to blank, IC50 was calculated in Prism using equation Y=Bottom + (Top-Bottom)/(1 + ((X^HillSlope)/(IC50^HillSlope))) and the standard deviation was calculated by multiplying standard error by square root of n (n being at least 3 replicates). To generate curves, for bacteria grown in absence of antibiotic (0 µM), since coordinates are plotted as logarithms and since log(0) is undefined, this point has been approximated 2 log units below the lowest tested value (*i.e.* 0.0625 µM) consistently with Prism user guide[78]. Curve fitting was performed with non-linear fitting function "log(inhibitor) vs. response -- Variable slope" using Y=Bottom + (Top-Bottom)/(1 + 10^((LogIC50-X)*HillSlope)). Both equations gave strictly the same result for $IC_{50}$.

### In vivo induction assay

A saturated preculture of *E. coli* DB10 transformed with pMMB plasmid was grown overnight at 37 °C under vigorous shaking in LB with 100 µg.ml⁻¹ ampicillin. For bacteria double-transformed with pMMB and pBAD plasmids, media was also supplemented with 20 µg.ml⁻¹ chloramphenicol and supplemented with 0.4% (w/v) Glucose. In vivo induction assay was performed in Mueller-Hinton broth (MH, Sigma Aldrich). A 96-wells flat-bottom plate (Flacon) was filled with a final volume per well of 200 µl, containing 100 µg.ml⁻¹ ampicillin, 1 mM Isopropyl β-D-1-thiogalactopyranoside (IPTG). Growth medium was supplemented with 20 µg.ml⁻¹ chloramphenicol and 0.2 % (w/v) L-Arabinose if pBAD plasmid is present. Induction of fluorescent reporters being antibiotic-dependent, growth media were supplemented with required antibiotic accordingly. Plates were therefore incubated in CLARIOstar Plus plate reader (BMG labtech) at 37 °C with 600 rpm double-orbital shaking up to 24 h, $OD_{600}$ and fluorescence (excitation: 497-15 nm, emission: 540-20 nm, gain 1600) being measured each 30 min.

### Immunoblotting

Samples were loaded on 12.5 % acrylamide SDS-PAGE gels. They were resolved by migrating at 0.04 A and then applied on Immun-Blot PVDF membrane (Bio-Rad) that had been activated in absolute ethanol then washed in electro-transfer buffer (25 mM Tris base, 192 mM glycine, 0.1% (w/v) SDS, 10% (v/v) absolute ethanol). Proteins were electro-transferred at 100 V during 1 h. Membranes were blocked by incubating 1 h in 1X PBS supplemented with 0.5% nonfat dry milk then washed with 1X PBS. C-term His-tagged MsrD variants were detected using anti-6xHis-tag primary antibody (Covalab, 1:2 000 dilution in 1X PBS, 0.1% Tween-20) combined with anti-mouse-HRP secondary antibody (Covalab, 1:20 000 dilution in 1X PBS, 0.1% Tween-20). Immunoblots were revealed by performing an ECL detection using Clarity Western ECL Substrate (Bio-Rad) and imaged with Li-Cor Odyssey FC imaging system (Li-Cor).

### Preparation of *E. coli* DB10 ribosomes

An overnight saturated *E. coli* DB10 preculture was used to inoculate growth medium (LB, dilution 1:100) and cells were grown at 37 °C and 180 rpm. When $OD_{600} = 0.5$, cells were washed and harvested by two centrifugations during 20 min, 5 000 xg at 4 °C followed by resuspensions in Buffer A (20 mM Tris pH 7.4, 10 mM Mg(OAc), 100 mM NH₄(OAc), 0.5 mM EDTA). Pellets were resuspended in Buffer

A supplemented with 6 mM β-mercaptoethanol, 25 µg.ml⁻¹ lysozyme, 0.001% (v/v) protease inhibitor cocktail (Sigma Aldrich) and lysed three times at 2.5 kBar using a cell disrupter (Constant Systems Limited). Lysate was clarified by two centrifugations during 15 min, 22 000 xg at 4 °C then spun for 20 h, 34 700 rpm at 4 °C in a Type 70 Ti rotor (Beckmann Coulter) through a 37.7% (w/v) sucrose cushion in Buffer B (20 mM Tris pH 7.4, 10 mM Mg(OAc), 500 mM NH₄(OAc), 0.5 mM EDTA, 6 mM β-mercaptoethanol). Sucrose cushions were decanted and ribosome pellets resuspended in Buffer C (20 mM Tris pH 7.4, 7.5 mM Mg(OAc), 60 mM NH₄(OAc), 0.5 mM EDTA, 6 mM β-mercaptoethanol). Finally, 12 mg of ribosomes were loaded onto 10-40 % (w/v) sucrose gradients in Buffer C and centrifuged 14 h at 20 000 rpm, 4 °C in SW28 rotor (Beckman Coulter). Gradients were fractionated using Biocomp Piston Gradient Fractionator (BioComp Instruments), absorbance being measured at 254 nm. Fractions containing 70 S ribosomes were pooled, washed and concentrated in Amicon 50k (Merckmillipore) using Buffer C. Ribosomes concentration was adjusted to 18-20 µM, then aliquoted and flash frozen in liquid nitrogen.

### RNA extraction and northern blotting

Total RNA extraction was realized using the RNAsnap method as previously described[79]. In brief, a preculture of *E. coli* DB10 containing pMMB-*msrDL-msrD*$_{(1-3)}$:*yfp* was grown overnight at 37 °C under vigorous shaking in LB supplemented with 100 µg.ml⁻¹ ampicillin. Growth medium (MH, 100 µg.ml⁻¹ ampicillin) was inoculated at $OD_{600} = 0.05$, and cells were grown at 37 °C and 180 rpm. When $OD_{600} = 0.5$, cells were treated (or not) with 1 mM IPTG during 10 min, then treated (or not) with 100 nM erythromycin. Samples of 2 ml were collected ($T_{0'}$ = before IPTG was added, $T_{10'}$ = 10 min after IPTG was added, and 5 min, 15 min, 30 min, 60 min and 120 min after erythromycin was added), then spun during 1 min at 15 000 rpm. Growth media was removed and cells were resuspended in 100 µl RNAsnap buffer (95% (v/v) formamide, 18 mM EDTA, 0.025% (v/v) SDS, 1% (v/v) β-mercaptoethanol). Samples were heated for 10 min at 95 °C then clarified by centrifuging 10 min, at 16 000 rpm at 16 °C. For northern blotting analysis, 6 µg of total RNAs were resolved on a 1% (w/v) agarose gel then transferred onto Amersham Hybond-N+ Membrane (GE Healthcare) by capillary transfer. Radioactive probe was prepared using 40 pmol of primer 59 and 5′-labeled with 10 U of T4 Polynucleotide Kinase (New England Biolabs) and [γ³²P]ATP (150 µCi). Probe was hybridized overnight at 42 °C using ULTRAhyb-Oligo hybridization buffer (Thermo Fisher Scientific). Membrane was washed three times at 42 °C during 15 min (once in 2x SSC + 0.1% (v/v) SDS, once in 1x SSC + 0.1% (v/v) SDS and finally in 0.1x SSC + 0.1% (v/v) SDS). Radioactive signal was visualized by exposing 4 h a Storage Phosphor Screen BAS-IP MS 2040 (Fujifilm) then imaged with a Typhoon FLA 9500 (GE Healthcare).

### In vitro transcription

In vitro transcription was carried out in T7 RiboMAX Large Scale RNA Production System kit (Promega) according to manufacturer instructions. Briefly, to generate DNA matrix (Supplementary Table 5), T7 promoter (5′-GCGAATTAATACGACTCACTATAGGG-3′) was added by PCR using primers pair 56/57 (Supplementary Table 4) and pMMB plasmids as templates. Transcription reactions were incubated 4 h at 37 °C and transcripts were purified with TRIzol Reagent (Thermo Fisher Scientific) and Direct-zol RNA Miniprep kit (Zymo Research), samples being eluted in THE Ambion RNA Storage Solution (Thermo Fisher Scientific). Final concentration was adjusted to 5 pmol.µl⁻¹.

### Toeprinting assay

Position of stalled ribosomes on mRNA was determined by toeprinting assay, slightly adapted from previously described methods[15,80]. In vitro translation reactions were performed using PURExpress ΔRF123 kit and PURExpress ΔRibosome (New England Biolabs) according to manufacturer instructions. Briefly, prior to in vitro translation

reactions, 0.5 µl of 500 µM ligand (e.g. erythromycin etc) was dried in a micro-centrifuge tube, using a SpeedVac vacuum concentrator (Thermo Fisher Scientific), final concentration being 50 µM once resuspended in a 5 µl reaction. Reactions were incubated during 15 min at 37 °C using 5 pmol of RNA templates (generated as described above) and 3.3 µM of purified *E. coli* DB10 ribosomes. When needed, samples were treated with 100 µM final puromycin and incubated 3 min more. Immediately, 2 pmol of CY5-labeled primer 58 complementary to NV1 sequence[20] was added and incubated for 5 min at 37 °C. For each reaction, reverse transcription was performed with 0.5 µl (corresponding to 5 U) of Avian Myeloblastosis Virus Reverse Transcriptase (Promega), 0.1 µl dNTP mix (10 mM), 0.4 µl Pure System Buffer (5 mM K-phosphate pH 7.3, 9 mM Mg(OAc), 95 mM K-glutamate, 5 mM NH$_4$Cl, 0.5 mM CaCl$_2$, 1 mM spermidine, 8 mM putrescine, 1 mM DTT) and incubated 20 min at 37 °C. Reactions were quenched with 1 µl NaOH 5 M and incubated 15 min at 37 °C. Alkali were therefore neutralized by adding 0.7 µl HCl 25% immediately supplemented with 20 µl toe-printing resuspension buffer (300 mM Na(OAc) pH 5.5, 5 mM EDTA, 0.5% (v/v) SDS). Finally, cDNAs were purified using QIAquick Nucleotide Removal kit (Qiagen), vacuum dried and resuspended in 6 µl formamide dye (95% (v/v) formamide, 20 mM EDTA, 0.25% (w/v) bromophenol blue). Sanger sequencing was performed on DNA matrix used for in vitro transcription. Briefly, each 20 µl reaction contained 7.5 nM DNA matrix, 75 nM CY5-labeled primer 58, 40 µM of each dNTPs, 0.025 U Taq Pol (New England Biolabds), 1x Thermo Pol Buffer (New England Biolabs) and corresponding ddNTPs (625 µM ddCTP/ ddTTP/ddATP or 50 µM ddGTP). After PCR, 20 µl formamide dye was added to each sequencing reaction. Samples were denatured 3 min at 80 °C, then 5 µl of sequencing reaction and 1.5 µl of toe-printing reaction were loaded on a 6% acrylamide/bis-acrylamide (19:1) sequencing gel containing 8 M urea. Gel was resolved by migrating 90 min at 50 W, then imaged using a Typhoon FLA 9500 (GE Healthcare Life Sciences) using CY5 mode, LPR Ch.2 filter and 635 nm laser.

### Preparation of MsrDL-SRC

A DNA matrix (Supplementary Table 5) was prepared with CloneAmp HiFi PCR Premix (Takara) using primer pair 56/60 (Supplementary Table 4) and plasmid pMMB-*msrDL-msrD*$_{(1-3)}$:*yfp* as matrix. The corresponding mRNA was generated as described above. The complete sequence of MsrDL-SRC mRNA is:

5′- GGAGCGGAUAACAAGAUACUGAGCACAACAAUAUU**GGAGG**A AUAUUUAUGUAUCUUAUUUUCAUGUAACUCUUCCUGCUAAAAUCG CAGGGUUUUCCCUGC −3′ (RBS, bold; *msrDL*, shaded gray; the ATG codon in the P site of stalled ribosomes is underlined). Prior to in vitro translation reaction, 1 µl of 500 µM erythromycin (50 µM final concentration once resuspended in 10 µl) has been dried in a micro-centrifuge tube, using a SpeedVac vacuum concentrator (Thermo Fisher Scientific). Dried erythromycin has been resuspended in a 10 µl in vitro translation reaction carried out in PURE*frex* 2.0 kit (Genefrontier), containing 1.83 µM of purified *E. coli* DB10 ribosomes and 3.6 µM mRNA (molar ratio 1:2). The reaction has been incubated for 10 minutes at 37 °C, before adding 100 µM final puromycin, and incubated 3 min more. Reaction was therefore diluted in ribosome purification Buffer C (20 mM Tris pH 7.4, 7.5 mM Mg(OAc), 60 mM NH$_4$(OAc), 0.5 mM EDTA, 6 mM β-mercaptoethanol) to reach a concentration of 150 nM ribosomes. Cryo-EM grids were immediately prepared.

### Cryo-EM grids preparation

Safematic ccu-010 HV carbon coater was used to coat Quantifoil carbon grids (QF-R2/2-Cu) with a thin carbon layer of approximate thickness of 2 nm. Grids were then glow discharged for 20 sec at 3 mA. Then, 4 µl of in vitro translation reaction diluted to 150 nM were applied, and after a 2 sec blotting (force 5) and 30 sec waiting time, grids were vitrified in liquid ethane using a Vitrobot Mark IV (FEI) set to 4 °C and 100 % humidity.

### Image acquisition and processing

Data collection was performed on a Talos Arctica instrument (FEI Company) at 200 kV using the SerialEM software for automated data acquisition. Data were collected at a nominal defocus of −0.5 to −2.7 µm at a magnification of 120,000 X yielding a calibrated pixel size of 1.2 Å. Micrographs were recorded as movie stack on a K2 Summit direct electron detector (Gatan), each movie stack were fractionated into 65 frames for a total exposure of 6.5 sec corresponding to an electron dose of 64 e$^-$/Å$^2$. MotionCor2[81] was used for dose weighting, drift and whole-frame motion correction. A dose weighted average image of the whole stack was used to determine the contrast transfer function with the software Gctf[82]. Particles were picked using a Laplacian of gaussian function (min diameter 260 Å, max diameter 320 Å). A total of ~254k particles were extracted from a subset of motion-corrected images (1523 micrographs) presenting a resolution equal or better than 4 Å with a box size of 360 pixels. The particles were binned three folds for 2D and subsequent 3D classification. After 2 rounds of 2D classification in RELION 3[83], ~158k particles were selected and submitted to Relion 3D classification[83]. A class of non-rotated 70 S depicting high-resolution features and bearing a tRNA with various occupancies was selected, representing ~104k particles. This class was further 3D-classified with a spherical mask engulfing the tRNAs binding sites (Supplementary Fig. 4b) into 4 classes, thus yielding to P- and E-tRNAs, P-tRNAs only, A- and P-tRNAs and E-tRNAs 70 S reconstructions. Only P- and E-tRNAs and P-tRNAs only reconstructions were used for further processing, which represents ~62k particles displaying the same global conformation of the 70 S. These particles where re-extracted at the full pixel size (1.2 Å) and refined through the 3D auto-refinement performed in RELION 3[83] resulting in a 3 Å reconstruction (Supplementary Fig. 4c), after CTF-refinement, Bayesian particle polishing and post-processing in RELION 3. In spite of the sufficient local resolution at the vicinity of MsrDL peptide at the PTC ( ~3 Å), residual movements of the 30 S around the 50 S can be deduced by the lower local resolutions of the head and the body of the 30 S. In order to improve their resolution so to derive a complete atomic model of the entire 70 S complex, we applied RELION 3 multi-body refinement by defining three bodies; 50 S, 30S-head and 30S-body. After completion of the refinement, the reconstructions of the 30S-head, 30S-body and 50 S reached average resolutions of 3.3, 3.08 and 2.97 Å, respectively. The local resolution (Supplementary Fig. 4b, d) was estimated using ResMap[84].

### Model building and refinement

The atomic model of erythromycin-stalled *Escherichia coli* 70 S ribosome with streptococcal MsrDL nascent chain was built into cryo-EM maps using Coot and Phenix[85,86]. Insofar as *E. coli* DB10 hasn't been sequenced yet, we assumed that ribosomal proteins and rRNAs were strictly identical to *E. coli* K12. Furthermore, we did not notice any significant features in the map. Structure of SpeFL-SRC in response to L-ornithine was used as initial model (PDB 6TC3)[15] and has been fitted in 70 S ribosome map EMD-13805. Then, each part of the ribosome (50 S, 30 S Body and 30 S Head) has been individually inspected in the corresponding map (respectively EMD-13806, EMD-13807 and EMD-13808) modified if necessary in Coot. P-site elongator tRNA $_{Met}$tRNA$^{Met}$ was modeled de novo based on *E. coli* K12 MG1655 *metT* gene, post-transcriptional modifications were added consistently with Modomics database[87] (http://genesilico.pl/modomics/). E-site tRNA $_{Phe}$tRNA$^{Phe}$ was derived from crystal structure of phenylalanine tRNA from *E. coli* (PDB 6Y3G) with minor adjustments[88]. ERY, MsrDL leader peptide and mRNA were modeled de novo in Coot. Final model was refined in map EMD-13805 using Phenix[86].

### Figure preparation

Growth curves and histograms were generated using GraphPad Prism 7 (GraphPad). Sequence alignments were visualized with JalView[89]. Western blotting, northern blotting and toe-printing gels were

analyzed using Fiji[90]. Figures depicting molecular structures or electronic density maps were prepared using PyMOL Molecular Graphics System, Chimera and ChimeraX[91,92].

## Statistics and reproducibility

Statistical details can be found in the figure legends. Statistical significance was assessed using unpaired two-tailed *t*-test function in Prism 7 (GraphPad) without adjustments. All the experiments presented have be reproduced at least twice with the same results. This includes all the Toeprinting experiments (Fig. 3a, c and e and supplementary Fig. 3d) and the Northern Blot (Fig. 2b).

## Reporting summary

Further information on research design is available in the Nature Portfolio Reporting Summary linked to this article.

## Data availability

The data that support this study are available from the corresponding authors upon reasonable request. Cryo-EM map of erythromycin-stalled *Escherichia coli* 70 S ribosome with streptococcal MsrDL nascent chain has been deposited at the Electron Microscopy Data Bank (EMDB) with accession code EMD-13805, as well as 50 S, 30 S Body and 30 S Head maps obtained after multibody refinement with accession code EMD-13806, EMD-13807, and EMD-13808, respectively. Corresponding atomic model has been deposited in the Protein Data Bank (PDB) with accession code 7Q4K. The *msrD* sequence is accessible in the Genebank database under No. FR671415. All the other sequences used for the sequence alignment in Supplementary Figs. 1 and 2 were extracted from the Genebank database, the reference Nos. are indicated in Supplementary Figure 2. Source data are provided with this paper.

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

## Acknowledgements

This work has benefited from the facilities and expertize of the Biophysical and Structural Chemistry platform (BPCS) at IECB, CNRS UMS3033, Inserm US001, University of Bordeaux and supported by funds from the CNRS (UMR8261) and Paris Cité University. C.R.F. is funded by a doctoral grant from the French Ministère de l'Enseignement supérieur, de la Recherche et de l'Innovation, F.O. and G.B received support from the LABEX program (DYNAMO ANR-11-LABX-0011), the ANR grants EZOtrad (ANR-14-ACHN-0027) for G.B. and ABC-F_AB (ANR-18-CE35-0010) for G.B. and Y.H., ERC-2017-STG #759120 "TransTryp" for Y.H.. E.C.L. and C.A.I. received funding for this project from the European Research Council (ERC) under the European Union's Horizon 2020 research and innovation program (Grant Agreement No. 724040). The authors would like to thank Dr. Olivier Chesneau (Département de Microbiologie, Institut Pasteur, Paris - France) for providing *E. coli* DB10 strain, plasmids pBAD33 and pVN50; Dr. Sylvain Durand and Dr. Maud Guillier (UMR 8261, CNRS, Université de Paris, Institut de Biologie Physico-Chimique, Paris - France) who provided bicyclomycin and equipment, respectively for toeprinting experiments. We also would like to thank Laura Monlezun and Tina Wang for proofreading the manuscript.

## Author contributions

F.O. performed the first experiments that initiated the project. C.R.F. performed most of the experiments following. S.N. performed northern blotting experiments. C.R.F. prepared the cryo-EM sample. H.S. prepared cryo-EM grids and imaged them. C.A.I. and Y.H. processed the cryo-EM data. C.R.F and E.C.L. reconstructed the atomic model. G.B. initiated the project and designed the research program with Y.H.. G.B. and C.R.F. wrote the paper with input from all authors. G.B., Y.H., S.N. and F.O. did the final manuscript revision.

## Competing interests

The authors declare no competing interests.
