## [Peer Review File · Nature Communications]

Regulation of the macrolide resistance ABC-F translation factor MsrDREVIEWER COMMENTS

Reviewer #1 (Remarks to the Author):

The manuscript of Fostier and coworkers addresses the regulation and mechanism of the antibiotic resistance ATP-binding cassette protein MsrD. These proteins have been shown to bind to drug-stalled ribosomes and confer resistance by dislodging the drug from its binding site. Despite recent insights from cryo-EM analysis there remains a lot of questions relating to their mechanism of action. Additionally, expression of these proteins is toxic, therefore, they are regulated in a drug-dependent manner using upstream open reading frames (uORFs). Here the authors dissect the regulation of MsrD showing that the expression is regulated by an upstream MsrDL uORF in a drug-dependent manner. The authors show that the macrolide erythromycin induces stalling on MsrDL with the stop codon in the A-site and this prevents transcription termination and thereby allows transcription and translation of the downstream MsrD resistance determinant. The highlight of the paper is a structure of the MsrDL-stalled ribosome, providing the mechanistic details as to how MsrDL and erythromycin interplay with the ribosome to prevent release factors from terminating translation. Although such systems are well known and there are already structures of such drug-dependent stalling complexes, the details for MsrD differ in the details. For me the insights into the regulation are well-performed, interpreted and nicely presented and represent a strength of the paper. By contrast, the weaknesses of the paper are the parts related to the mechanism of action of MsrD to confer resistance. The authors make some provocative claims that MsrD splits ribosomes but the evidence is weak and not at all convincing, further data supporting their claims would significantly improve this part of the paper – perhaps its simply better to remove it from the manuscript since its any not really needed for the regulation section?

Specific comments.

1. Figure 1c,d: It does appear that there is more MsrDWT migrating with the 50S fraction compared to the 70S fraction in the erythromycin-treated cells but its unclear how everything is quantified and normalized here, and whether these differences are really significant? It would be good to Western for ribosomal proteins to normalize the level of co-migration with small and large subunit. In this regard, how reproducible are the similarities and differences between the different constructs with and without erythromycin – both in terms of the profile as well as factor migration? Since the authors want to make strong statements on some of these differences, it would seem appropriate to provide multiple traces and demonstrate statistical significance.
2. The model that MsrD binds to the Ery-stalled ribosomes and splits them is exciting, however, the rationale for this model appears to be solely based on the observed association of MsrD with the 50S subunit. However, its not clear to me how the authors exclude the possibility that MsrD simply binds directly to 50S subunits - or whatever else is in the 50S peak – assembly intermediates. It would seem pertinent to back up these in vivo observations with in vitro experiments to validate the model.
3. Does MsrD split translating ribosomes and/or vacant 70S? Would not one expect therefore a dramatic reduction in polysomes and 70S ribosomes and a concomitant accumulation of 30S and 50S subunits? This does not appear to be the case, the polysomes go down and the 70S peak goes up (and 50S stays the same) - although neither very dramatically, suggesting conversion of polysomes to 70S ribosomes – this does not make the suggested model not very convincing. As mentioned above, it would seem pertinent to back up these in vivo observations with in vitro experiments to support the suggested model.
4. The disomes observed in the presence of ERY are suggested to arise due to elongation inhibition. Are they also observed with other elongation inhibitors, such as tetracycline or chloramphenicol? It would be a nice negative control to show that they remain upon MsrD expression.
5. Despite lacking validation, I don't really understand the logic of the model. How would splitting a ribosome help with the resistance problem? If each drug release event is followed by splitting, then no drug-stalled ribosomes will ever be able to finish translating their proteins- how will this help the cell to overcome the drug? Also what happens to the peptidyl-tRNA? Does it dissociate? Can the authors provide evidence for this also? If it remains on the 50S, then it should be the subject to some sort of

quality control system...can the authors provide any evidence for this? The few experiments presented in the paper open many more questions than they answer. Maybe they should be simply removed from the paper altogether or additional experiments should be performed to solidify model.

Reviewer #2 (Remarks to the Author):

The work of Fostier et al. describes genetic, biochemical and structural approaches aimed to characterize the regulation of expression of the MsrD protein of the ARE ABC-F group and its function/mechanism for conferring resistance to macrolide antibiotics. The manuscript is organized into three main sections, each of them with its own conclusions. 1) MsrD rescues Ery-stalled ribosomes leading to resumption of translation and cell survival. Authors suggest that the two ATP binding sites of MsrD differentially affect the association/dissociation of the protein to and from the ribosome and therefore play distinctive roles for the rescue mechanism. 2) Ery cause translation termination stalling of the *msrDL* regulatory leader ORF preceding the *msrD* resistance gene. Ribosome stalling at the last sense codon of *msrDL* prevents transcription termination, leading to *msrD* activation. Cryo-EM structure of the Ery-MsrDL-ribosome complex reveals that a cladinose-containing antibiotic forces a highly unique path of the MsrDL nascent peptide that causes conformational changes of the peptidyl transferase center unfavorable for proper accommodation of RF1/RF2 and hence, inhibition of peptide hydrolysis. 3) Expression of MsrD fulfills not only the function to confer macrolide resistance but also represses its own expression by dislodging the stalled ribosome complex on *msrDL* which, presumably, leads to reactivation of the transcriptional attenuation. The authors make a good effort to put together a well-rounded story to better explain the mechanisms of this important family of antibiotic resistant proteins but, unfortunately, the experimental data for parts 1 and 3 are too fuzzy and difficult to interpret. Part 2 is more solid but it also has room for improvement.

The following are just a few points of criticism:

- The effects of the MsrD variants, wt or mutant, are difficult to assess because:
 - no data on the level of expression for each of them are presented.
 - the plasmid-born expression of all of them is toxic for *E. coli* cells (and effects are shown for only a single concentration of Ery, for a specific time point, or for time courses that are too short), which complicates interpreting the data on resistance.
 - differential association of variants with 50S/70S/disomes cannot be accurately determined since the Western blots shown for the polysome profiles lack normalizing controls (i. e., WBs for a 50S ribosomal protein).
 - the model proposed for the mechanism of action of MsrD does not seem to differ from what has been proposed for other members of the ARE-ABC-F family. For the reasons mentioned above, the impact on the mechanism of the proposed asymmetry of the ATP binding sites is not solidly supported by the experimental data.

- The genetic, biochemical and structural data of the MsrDL stalled complex are convincing and well presented, however:
 - to better appreciate the Ery induction of the reporter I'd suggest to re-plot the data of Fig. 2a as Fluorescence/OD600 vs. Ery concentration
 - as the authors mention, the *msrD(1-3):yfp* reporter can be used to INDIRECTLY test the effect of amino acid mutations of MsrDL on the formation of the stalled complex. A much direct way, that would eliminate the possibility that the mutations could alter elements necessary for the induction (not related to the formation of the stalled complex), would be to test these mutations in the toeprinting assay.

Other points:

-Authors mention that Erm leaders (B, C, D) only act via inhibition of elongation. However, to my knowledge, inhibition of translation termination for these complexes has not been explored.

-To understand the information from the plots shown in Fig. 6, rather than the data of a single time point, time courses need to be shown.

-Ref 41 does not mention sensitivity of DB1 strain to macrolides; reasons for the sensitivity are unclear. Authors should provide the genotype of this strain (and of the parental PR7 strain) in Supp. Table 3

-label the disome peaks in panels c, d of Fig. 1

-The Western blot for the solubility assay shown in Supp. Fig. 1e is unclear. Why are there two bands, one above 50 and another under 25 kDa? In addition, since there is no normalization control, no conclusions can be made regarding the total expression level or the fraction of protein in the supernatant or pellet.

-In the legend to Supp Fig 1: "Species where msrD was found to disseminate alone with msrDL are indicated by an asterisk"-change alone for along.

- Page 16, lane 415 (Site I (not Site II), MsrDE125Q).

Reviewer #3 (Remarks to the Author):

In this manuscript, the authors address how MsrD, a streptococcal ABC-F family protein that confers macrolide resistance is regulated by its leader peptide MsrDL. Using *E. coli* DB10 strain, the authors first demonstrated that the expression of MsrD confers erythromycin-, azithromycin-, and telithromycin-resistances in DB10. The authors then performed polysome fractionation to observe the effect of the antibiotics and MsrD on the polysome profiling and behavior of wildtype and mutant derivatives of MsrD. Based on the results obtained, the authors propose a model in which MsrD binds to the ERY-stalled ribosomes and triggers dissociation of the 70S ribosome into the 50S and 30S subunits in an ATP-dependent manner to rescue the stalled ribosomes. Authors also performed in vivo and in vitro experiments to demonstrate that the msrD is regulated by the transcriptional attenuation mechanism, in which the drug-dependent ribosome stalling of MsrDL precludes premature transcription termination, leading to the induction of msrD. Cryo-EM structure of MsrDL-SRC revealed that the MsrDL nascent chain adopted a distinct hook-like conformation in the exit tunnel without interacting with erythromycin. The structural study suggests that the MsrDL stabilizes an uninduced conformation of the A-site of the ribosome such that it avoids the A-site accommodation of the release factor. Finally, the authors demonstrated that the expression of the msrD is negatively regulated by its own.

Major comments:

Although this study contains several significant results, I think the manuscript also contains overinterpretations of results that should be more carefully written or supported by additional experiments.

(1) Based on the results of the polysome profiling (Figure 1 and Supplementary Figure 1), the authors suggest that wildtype MsrD interacts more with the 50S subunit. In contrast, the MsrD(EQ2) mutant interacts more with the 70S ribosome. However, because the Western blotting signals of MsrD variants seem to spread throughout almost all the fractions, it is not convincing to me that the results

support the authors' interpretation and their model that MsrD binds to the drug-bound 70S ribosome and triggers the subunit dissociation in an ATP-dependent mechanism (P6. lines 140-143). If the authors would like to strengthen their conclusion, they should do additional experiments such as reconstitution of the ribosome splitting biochemically or pull-down experiments of MsrD(WT) and (EQ2) to test if WT precipitates more 50S subunit than EQ2 while EQ2 precipitate more 70S than WT. (2) The authors complain that MsrD(EQ2) is more soluble than MsrD(WT) based on the results shown in Supplementary fig 1c and 1e. There are many background signals in the pellet fraction of the wildtype but not that of the EQ2 variant (Supplementary fig 1e), possibly because more proteins were loaded for wildtype. Thus, authors should show internal control. In addition, there are two major bands (around 50 and 25 kDa) in Supplementary Fig 1c. Authors should indicate which represents MsrD.

(3) Page 7, lines 158-159: I could not follow the logic that led the authors to conclude that "interaction of MsrD with the 50S is necessary for antibiotic resistance mechanism".

(4) Page 14, lines 367-: Based on the results obtained using the fluorescent reporter, the authors propose that MsrD rescues MsrDL-SRC. However, the reporter assay should be more carefully interpreted because erythromycin is a translation inhibitor. Therefore, I think more direct evidence is required to demonstrate that MsrD rescues MsrDL-SRC.

Minor comments:

Page 5, line 112: I think 'ribosome disomes' should be 'ribosome dimers, or disomes'.

Page 4, line 130: 'western-blotting' should be 'western blotting'

Page 11, line 285: I could not understand the following sentence implies, "When both are uncoupled like in the MsrDL(7A-iso) construct the regulation is less efficient"

Page 12, line 292: "P-tRNAs" should be "P-tRNAs"

Paris, Thursday, February 16, 2023

General response to the reviewers:

We are grateful to the reviewers for the insightful points raised on how to improve our manuscript. It is our understanding that all the reviewers found our claim on the possible mode of action of MsrD challenging and not well supported. We acknowledge that point and based on the suggestion of reviewer #1, we have removed this part of the manuscript to keep the focus on the regulation of MsrD, since this part has been appreciated by all the reviewers. This modification led to the removal of the polysomes analysis (Figure 1c and d and Supplementary Figure 1c, d and e), our model of MsrD action (Figure 6, right side) and the text associated with those figures.

Consequently, we have concentrated our revision on the regulation mechanism by performing toeprinting assays on mutants of MsrDL to strengthen the conclusions of the alanine scanning and we have added these results to the revised manuscript (Supplementary Figure 2d). We also conducted several *in vitro* translation assays aimed at expressing *msrD* from its mRNA in presence of *msrDL* mRNA, but unfortunately due to the aggregation prone propriety of MsrD, the quality of these assays were insufficient to be presented in this revised version of the manuscript.

To increase the clarity of the results presented in the manuscript, we have provided more information. For the MCI figure presented in Figure 1 (now 1c was 1e, f, g) the full growth curves are presented in Supplementary Figure 1e. We now present a Western Blot, in Supplementary Figure 1c, that shows the expression level of all the *msrD* variants presented in this study.

The data presented in Figure 2b has been replotted as Fluorescence/OD₆₀₀ versus ERY concentration and is presented in Supplementary Figure 2b. We have added a panel (c) in Supplementary Figure 6 where the full growth curves used for Figure 6a are presented. We also present similar data to that presented in Figure 6a, but with different ERY concentrations, in Supplementary Figure 6a and b.

We have substantially revised the text of our revised manuscript to improve the clarity and address the reviewers' comments. We have uploaded a version of the manuscript with changes highlighted using Microsoft Word track changes option.

We would like to reiterate our appreciation for the insightful work done by the reviewers. The comments have led to substantial improvements in both the rigor and the clarity of our manuscript.

Point-by-point responses to reviews of NCOMMS-22-30572 by Fostier *et al.* (reviewers 1- 3):

• **Reviewer #1 (Remarks to the Author):**

The manuscript of Fostier and coworkers addresses the regulation and mechanism of the antibiotic resistance ATP-binding cassette protein MsrD. These proteins have been shown to bind to drug-stalled ribosomes and confer resistance by dislodging the drug from its binding site. Despite recent insights from cryo-EM analysis there remains a lot of questions relating to their mechanism of action. Additionally, expression of these proteins is toxic, therefore, they are regulated in a drug-dependent manner using upstream open reading frames (uORFs). Here the authors dissect the regulation of MsrD showing that the expression is regulated by an upstream MsrDL uORF in a drug-dependent manner. The authors show that the macrolide erythromycin induces stalling on MsrDL with the stop codon in

the A-site and this prevents transcription termination and thereby allows transcription and translation of the downstream MsrD resistance determinant. The highlight of the paper is a structure of the MsrDL-stalled ribosome, providing the mechanistic details as to how MsrDL and erythromycin interplay with the ribosome to prevent release factors from terminating translation. Although such systems are well known and there are already structures of such drug-dependent stalling complexes, the details for MsrD differ in the details.

For me the insights into the regulation are well-performed, interpreted and nicely presented and represent a strength of the paper. By contrast, the weaknesses of the paper are the parts related to the mechanism of action of MsrD to confer resistance. The authors make some provocative claims that MsrD splits ribosomes but the evidence is weak and not at all convincing, further data supporting their claims would significantly improve this part of the paper – perhaps it's simply better to remove it from the manuscript since it's any not really needed for the regulation section?

Authors' main response to Reviewer #1: We thank the Reviewer for his insightful and critical judgment on our work. We agree on the fact that the highlight of the manuscript is the regulation of *msrD* expression by the leader peptide MsrDL and we appreciate that the reviewer thinks that this part is well performed. We thank him for helping us realize that the part of the manuscript related to the mode of action of MsrD is too preliminary. We were originally thinking that those results may be interesting for the community because they suggested another kind of mechanism, but we understand there is not sufficient results for a solid claim and we did not anticipate the provocative aspect of this claim. Therefore, in agreement with the Nature Communication's editor, we followed the reviewer's advice and removed this part from the manuscript.

Specific comments.

- 1. Figure 1c,d: It does appear that there is more MsrDWT migrating with the 50S fraction compared to the 70S fraction in the erythromycin-treated cells but it's unclear how everything is quantified and normalized here, and whether these differences are really significant? It would be good to Western for ribosomal proteins to normalize the level of co-migration with small and large subunit. In this regard, how reproducible are the similarities and differences between the different constructs with and without erythromycin – both in terms of the profile as well as factor migration? Since the authors want to make strong statements on some of these differences, it would seem appropriate to provide multiple traces and demonstrate statistical significance.

- 2. The model that MsrD binds to the Ery-stalled ribosomes and splits them is exciting, however, the rationale for this model appears to be solely based on the observed association of MsrD with the 50S subunit. However, it's not clear to me how the authors exclude the possibility that MsrD simply binds directly to 50S subunits - or whatever else is in the 50S peak – assembly intermediates. It would seem pertinent to back up these in vivo observations with in vitro experiments to validate the model.

- 3. Does MsrD split translating ribosomes and/or vacant 70S? Would not one expect therefore a dramatic reduction in polysomes and 70S ribosomes and a concomitant accumulation of 30S and 50S subunits? This does not appear to be the case, the polysomes go down and the 70S peak goes up (and 50S stays the same) - although neither very dramatically, suggesting conversion of polysomes to 70S ribosomes – this does not make the suggested model not very convincing. As mentioned above, it would seem pertinent to back up these in vivo observations with in vitro experiments to support the suggested model.

- 4. The disomes observed in the presence of ERY are suggested to arise due to elongation inhibition. Are they also observed with other elongation inhibitors, such as tetracycline or chloramphenicol? It would be a nice negative control to show that they remain upon MsrD expression.

- 5. Despite lacking validation, I don't really understand the logic of the model. How would splitting a ribosome help with the resistance problem? If each drug release event is followed by splitting, then no drug-stalled ribosomes will ever be able to finish translating their proteins- how will this help the

cell to overcome the drug? Also what happens to the peptidyl-tRNA? Does it dissociate? Can the authors provide evidence for this also? If it remains on the 50S, then it should be the subject to some sort of quality control system...can the authors provide any evidence for this? The few experiments presented in the paper open many more questions than they answer. Maybe they should be simply removed from the paper altogether or additional experiments should be performed to solidify model.

Authors' response: We thank the reviewer for raising many points of criticism. The part concerning these experiments has been removed from the manuscript. We thank the reviewer for suggesting additional important experiments to solidify our early observations and we will build on it for our future research. Regarding point 5, we hypothesized a model where the toxicity of the antibiotic wasn't due at first to full inhibition of all protein synthesis, but by accumulation of stalled ribosome on mRNA at specific sites like the identified macrolide arrest motif "+X+" (Beckert, B. *et al.*, Nat Commun, 2021, 12, 4466; Ramu, H., *et al.*, 2009, Mol Microbiol 71, 811–824; Sothiselvam, *et al.*, Proc Natl Acad Sci U S A, 2014, 111, 9804–9809). This can lead to two different consequences: toxicity of those stalled complexes and/or depletion of the free ribosome by the formation of those stalled complexes. After dissociation the dissociated complexes can be recycled either by direct dissociation of the peptidyl-tRNA by MrsD or with the help of other factors. We think that the reviewer raised a critical point that cannot be addressed in the revision of the current manuscript because of the extent of the needed experiments in order to unambiguously demonstrate our previously proposed model. In the future, we will attempt to address this point in a separate research project.

- **Reviewer #2 (Remarks to the Author):**

The work of Fostier et al. describes genetic, biochemical and structural approaches aimed to characterize the regulation of expression of the MsrD protein of the ARE ABC-F group and its function/mechanism for conferring resistance to macrolide antibiotics. The manuscript is organized into three main sections, each of them with its own conclusions. 1) MsrD rescues Ery-stalled ribosomes leading to resumption of translation and cell survival. Authors suggest that the two ATP binding sites of MsrD differentially affect the association/dissociation of the protein to and from the ribosome and therefore play distinctive roles for the rescue mechanism. 2) Ery cause translation termination stalling of the msrDL regulatory leader ORF preceding the msrD resistance gene. Ribosome stalling at the last sense codon of msrDL prevents transcription termination, leading to msrD activation. Cryo-EM structure of the Ery-MsrDL-ribosome complex reveals that a cladinose-containing antibiotic forces a highly unique path of the MsrDL nascent peptide that causes conformational changes of the peptidyl transferase center unfavorable for proper accommodation of RF1/RF2 and hence, inhibition of peptide hydrolysis. 3) Expression of MsrD fulfills not only the function to confer macrolide resistance but also represses its own expression by dislodging the stalled ribosome complex on msrDL which, presumably, leads to reactivation of the transcriptional attenuation. The authors make a good effort to put together a well-rounded story to better explain the mechanisms of this important family of antibiotic resistant proteins but, unfortunately, the experimental data for parts 1 and 3 are too fuzzy and difficult to interpret. Part 2 is more solid but it also has room for improvement.

Authors' main response to Reviewer #2: We thank the reviewer for this input and judgment on our work and we appreciated that the reviewer found our story well-rounded. As explained in the general and reviewer #1 responses, we have decided to remove part 1 of the manuscript, as we realized that it brings some confusion and many more experiments need to be performed to demonstrate in a clear and solid manner the mechanism of action of MsrD. We hope that the new toeprint results and extended data that we have added, along with the revision of the main text, will make parts 2 and 3 clearer and more solid for the reviewer.

The following are just a few points of criticism:

- The effects of the MsrD variants, wt or mutant, are difficult to assess because:
 - no data on the level of expression for each of them are presented.

Authors' response: We thank the reviewer for raising this point. A western blot showing the expression of each MsrD variant is now presented in Supplementary Figure 1c. This western blot shows that most of the mutants express similarly to the WT and that the mutants that reduce MsrD ATPase activity express less, as expected due to their increase toxicity for the cell.

-the plasmid-born expression of all of them is toxic for E. coli cells (and effects are shown for only a single concentration of Ery, for a specific time point, or for time courses that are too short), which complicates interpreting the data on resistance.

Authors' response: There is some confusion here, we performed all the experiments, except Fig 6, at different concentrations of Ery and followed the effect in a time resolve manner. Figure 1c shows the effect for all the mutants at different Ery concentrations (from 0 to 64 μ M), but in our initial submission we didn't add the raw data which shows the OD over time. Guided by the reviewer, we now have added the raw data corresponding to Figure 1c in the Supplementary Figure 1e. The time courses are done for 24h which we think is enough. For Figure 6, we now added a similar experiment, but for higher concentrations of ERY (500 nM and 1000 nM) for some of the mutants.

-differential association of variants with 50S/70S/disomes cannot be accurately determined since the Western blots shown for the polysome profiles lack normalizing controls (i. e., WBs for a 50S ribosomal protein).

Authors' response: We thank the reviewer for raising this point. The part concerning these experiments has been removed from the manuscript as explained above.

-the model proposed for the mechanism of action of MsrD does not seem to differ from what has been proposed for other members of the ARE-ABC-F family. For the reasons mentioned above, the impact on the mechanism of the proposed asymmetry of the ATP binding sites is not solidly supported by the experimental data.

Authors' response: The exact mechanism of action of the ARE ABC-F protein is not understood and most of the existing models are based on the structure of ATPase dead mutant or with non-hydrolyzable ATP analogue. Initially, we believed that our data would bring new and insightful information about the possible mechanism of action of this ARE ABC-F, which we have now reconsidered in light of the thoughtful reviewers' comments. We acknowledge that our data falls short in justifying our proposed model of alternative mechanism of action that was presented in our first submission of this manuscript, which reviewer #1 described as provocative. We have therefore removed this part from the manuscript.

- The genetic, biochemical and structural data of the MsrDL stalled complex are convincing and well presented, however:

-to better appreciate the Ery induction of the reporter I'd suggest to re-plot the data of Fig. 2a as Fluorescence/OD600 vs. Ery concentration.

Authors' response: We appreciate the reviewer finding this part convincing. We plotted the OD600 vs Fluorescence intensity as a classical way in bacteria physiology to assess change in fluorescence for cultures that do not grow at the same speed. We understand the comment of the reviewer and have now added the requested plot in Supplementary Figure 2b.

- as the authors mention, the *msrD(1-3):yfp* reporter can be used to INDIRECTLY test the effect of amino acid mutations of MsrDL on the formation of the stalled complex. A much direct way, that would eliminate the possibility that the mutations could alter elements necessary for the induction (not related to the formation of the stalled complex), would be to test these mutations in the toeprinting assay.

Authors' response: We thanks the reviewer to have guided us in this direction. We have performed new toeprint assays with the two variants of the MsrDL that have shown the strongest effect on the alanine scan, namely MsrDL_{L3A} and MsrDL_{L4A}. The toeprint assays, presented in Supplementary figure 3d, show that both constructs have less MsrDL-SRC formation in the presence of ERY and that the stalled complexes were not resistant to puromycin. These results confirm that the mutations L3A and L4A of MsrDL reduce the formation ERY-induced stalled ribosome and that the remaining stalled ribosomes have their PTC functional as opposed to the MsrDL-SRC on the WT-MsrDL which has its PTC silenced. These results also support a good agreement between the formation of MsrDL-SRC and the fluorescent signal of the *msrD(1-3):yfp* reporter.

Other points:

-Authors mention that Erm leaders (B, C, D) only act via inhibition of elongation. However, to my knowledge, inhibition of translation termination for these complexes has not been explored.

Authors' response: These leaders employ macrolide arrest motifs "+X+" (Beckert, B. *et al.*, Nat Commun, 2021, 12, 4466; Ramu, H., *et al.*, 2009, Mol Microbiol 71, 811–824; Sothiselvam, *et al.*, Proc Natl Acad Sci U S A, 2014, 111, 9804–9809) to mediate translational arrest during the elongation. Nonetheless in a publication concerning ErmCL (Ramu *et al.*, 2011, Molecular Cell. Supplementary figure 3), the authors replaced the sense codon just after the motif by a stop codon, and this ErmCL variant was able to stall at the termination as well.

-To understand the information from the plots shown in Fig. 6, rather than the data of a single time point, time courses need to be shown.

Authors' response: We apology for not presenting the time courses, they are now presented in Supplementary Figure 6c. In fact, in Figure 6a all points are measured at 30min intervals.

-Ref 41 does not mention sensitivity of DB1 strain to macrolides; reasons for the sensitivity are unclear. Authors should provide the genotype of this strain (and of the parental PR7 strain) in Supp. Table 3.

Authors' response: We apology for not including this information. We agree on the fact that the reference concerning the *E. coli* DB10 strain (Datta *et al.*, 1974. J. Gen. Microbiol.) is not explicit. However, this strain has been used in other publications concerning erythromycin resistance (For example: Arthur *et al.*, 1986. ASM. DOI: <https://doi.org/10.1128%2Faac.30.5.694>; Chesneau *et al.*, 2006. FEMS Microbiol. Lett. DOI: <https://doi.org/10.1111/j.1574-6968.2007.00643.x>). The first publication indicates the genotype (*thiA leu rna pnp gyrA rpsL*) and both publications indicate a sensitivity to macrolides. Consequently, we added the genotype and sensitivity of the strain in Supplementary table 3 as well as the reference Arthur *et al.*, 1986. ASM.

-label the disome peaks in panels c, d of Fig. 1

Authors' response: We thank the reviewer for spotting this typo. The part concerning these experiments has been removed from the manuscript as explained above.

-The Western blot for the solubility assay shown in Supp. Fig. 1e is unclear. Why are there two bands, one above 50 and another under 25 kDa? In addition, since there is no normalization control, no conclusions can be made regarding the total expression level or the fraction of protein in the supernatant or pellet.

Authors' response: We thank the reviewer for this fine observation. We think it's probably a proteolysis product. The part concerning these experiments has been removed from the manuscript.

-In the legend to Supp Fig 1: "Species where *msrD* was found to disseminate alone with *msrDL* are indicated by an asterisk"-change alone for along.

Authors' response: We thank the reviewer for spotting this typo. The sentence has been rephrased.

- Page 16, lane 415 '(Site I (not Site II), MsrDE125Q).

Authors' response: We thank the reviewer for spotting this error. The part concerning these experiments has been removed from the manuscript.

- **Reviewer #3 (Remarks to the Author):**

In this manuscript, the authors address how MsrD, a streptococcal ABC-F family protein that confers macrolide resistance is regulated by its leader peptide MsrDL. Using *E. coli* DB10 strain, the authors first demonstrated that the expression of MsrD confers erythromycin-, azithromycin-, and telithromycin-resistances in DB10. The authors then performed polysome fractionation to observe the effect of the antibiotics and MsrD on the polysome profiling and behavior of wildtype and mutant derivatives of MsrD. Based on the results obtained, the authors propose a model in which MsrD binds to the ERY-stalled ribosomes and triggers dissociation of the 70S ribosome into the 50S and 30S subunits in an ATP-dependent manner to rescue the stalled ribosomes. Authors also performed in vivo and in vitro experiments to demonstrate that the *msrD* is regulated by the transcriptional attenuation mechanism, in which the drug-dependent ribosome stalling of MsrDL precludes premature transcription termination, leading to the induction of *msrD*. Cryo-EM structure of MsrDL-SRC revealed that the MsrDL nascent chain adopted a distinct hook-like conformation in the exit tunnel without interacting with erythromycin. The structural study suggests that the MsrDL stabilizes an uninduced conformation of the A-site of the ribosome such that it avoids the A-site accommodation of the release factor. Finally, the authors demonstrated that the expression of the *msrD* is negatively regulated by its own.

Authors' main response to Reviewer #3: We thank the reviewer for reviewing our manuscript, his/her suggestions and input. As explained in the general and the two previous reviewers responses, guided by the comments of all 3 reviewers, who think that the part of the manuscript concerning the mechanism of action of MsrD is too preliminary. We have followed the suggestion of reviewer #1 and removed this part for our manuscript. We have focused our revision on the regulation of *msrD* expression.

Major comments:

Although this study contains several significant results, I think the manuscript also contains overinterpretations of results that should be more carefully written or supported by additional experiments.

- (1) Based on the results of the polysome profiling (Figure 1 and Supplementary Figure 1), the authors suggest that wildtype MsrD interacts more with the 50S subunit. In contrast, the MsrD(EQ2) mutant interacts more with the 70S ribosome. However, because the Western blotting signals of MsrD variants seem to spread throughout almost all the fractions, it is not convincing to me that the results support the authors' interpretation and their model that MsrD binds to the drug-bound 70S ribosome and triggers the subunit dissociation in an ATP-dependent mechanism (P6. lines 140-143). If the authors would like to strengthen their conclusion, they should do additional experiments such as reconstitution of the ribosome splitting biochemically or pull-down experiments of MsrD(WT) and (EQ2) to test if WT precipitates more 50S subunit than EQ2 while EQ2 precipitate more 70S than WT.

- (2) The authors complain that MsrD(EQ2) is more soluble than MsrD(WT) based on the results shown in Supplementary fig 1c and 1e. There are many background signals in the pellet fraction of the wildtype but not that of the EQ2 variant (Supplementary fig 1e), possibly because more proteins were loaded for wildtype. Thus, authors should show internal control. In addition, there are two major bands (around 50 and 25 kDa) in Supplementary Fig 1c. Authors should indicate which represents MsrD.

- (3) Page 7, lines 158-159: I could not follow the logic that led the authors to conclude that "interaction of MsrD with the 50S is necessary for antibiotic resistance mechanism".

Authors' response: We appreciate that the reviewer thinks that our study contains significant results. As explained above, we acknowledge that we have presented some results that remain, at this stage, a bit too preliminarily. We initially believed that these results may be of interest to the community because they suggested an alternative mechanism of action for MsrD. However, thanks to reviewers' comments, we understand that more experiments are needed to back our model. Therefore, in agreement with the Editor, we followed reviewer #1's advice and have removed this part from the manuscript. We will use these comments to guide our future experiments in this aspect of our research. The attempt to perform *in vitro* ribosome splitting experiments were hampered by the lack of solubility of the purified MsrD. The pulldown is a good suggestion, we will try it in our future work. Concerning the solubility test (point 1), the number of cells were normalized before being loaded on the gel, but because the mutant EQ2 creates a strong toxicity for the cell, it is less expressed. Nevertheless, we agree that these experiments will need more internal control. For point (2), the 50 kDa band is MsrD, the 25 kDa one is a degradation product often visible with ABC-F proteins. For point (3), this statement came from the fact that MsrDE125Q variant provides some level of antibiotic resistance and like the WT MsrD they both show more interaction with the 50S on the polysome than the EQ2 and E434Q variants.

- (4) Page 14, lines 367-: Based on the results obtained using the fluorescent reporter, the authors propose that MsrD rescues MsrDL-SRC. However, the reporter assay should be more carefully interpreted because erythromycin is a translation inhibitor. Therefore, I think more direct evidence is required to demonstrate that MsrD rescues MsrDL-SRC.

Authors' response: We thank the reviewer for raising this point. However, similar results were obtained for the ARE ABC-F Vga(A) using the same experimental approach (See Figure 3 in Vimberg *et al.*, 2020. ASM. DOI: <https://doi.org/10.1128/AAC.00666-20>). As we explained in the manuscript, like most ARE ABC-F, the protein MsrD is prone to aggregation and we have unsuccessfully applied a plethora of protocols to obtain purified soluble samples of MsrD. Therefore, the *in vivo* tests proposed by the reviewers with purified MsrD can unfortunately not be performed in an acceptable timeline for

our revision, and will require substantial efforts that go beyond the scope of this particular project. But we have performed new toeprint assays on MsrDL mutant, presented in Supplementary Figure 3d. These results confirm that the mutations L3A and I4A of MsrDL reduce the formation ERY-induced stalled ribosome and that the remaining stalled ribosomes have their PTC functional as opposed to the MsrDL-SRC on the WT-MsrDL which have its PTC silenced. These results also support a good agreement between the formation of MsrDL-SRC and the fluorescent signal of the *msrD(1-3):yfp* reporter. Nevertheless, we agree with the reviewer that this result doesn't prove a direct function of MsrD on the MsrDL-SRC, which we state in the manuscript: "If MsrD is also able to rescue this SRC, it will repress its own expression and form a feedback loop similarly to *Vga(A)*".

Minor comments:

- Page 5, line 112: I think 'ribosome disomes' should be 'ribosome dimers, or disomes'.

Author's response: We thank the reviewer for raising this point. The part concerning these experiments has been removed from the manuscript.

- Page 4, line 130: 'western-blotting' should be 'western blotting'.

Authors' response: We thank the reviewer for spotting this typo. The part concerning these experiments has been removed from the manuscript.

- Page 11, line 285: I could not understand the following sentence implies, "When both are uncoupled like in the MsrDL(7A-iso) construct the regulation is less efficient".

Authors' response: We thank the reviewer for raising this point. The sentence has been rephrased.

- Page 12, line 292: "P-tRNAs" should be "P-tRNAs".

Authors' response: We thank the reviewer for spotting this typo. It has been corrected.

Grégory Boël, PhD
Principal Investigator
CNRS / Université de Paris, UMR8261
Institut de Biologie Physico-Chimique
13 rue Pierre et Marie Curie, 75005
Paris, France

Yaser Hashem, PhD
Principal Investigator
INSERM U1212 (ARNA)
Institut Européen de Chimie et Biologie
Université de Bordeaux, 33607 Pessac,
France

REVIEWERS' COMMENTS

Reviewer #1 (Remarks to the Author):

the authors have removed the section of the manuscript to which all my comments were addressed - I think this makes the manuscript much more solid, but I hope the authors will continue to follow-up their splitting mechanism for ABCF, this would be very interesting for the community if it could be validated.

Reviewer #2 (Remarks to the Author):

Comments on the revised manuscript

Main criticism:

Authors have made a great effort in improving the manuscript. Removing the section of the MsrD mode of action was a good decision. Including the in vitro toeprinting analysis of the selected MsrDL mutants further enhances the analysis of the nascent peptide and antibiotic mediated stalling which, in my opinion, remains the most solid and somehow novel portion of the work. However, the inducible expression of msrD by releasing transcriptional attenuation lacks novelty. Similarly, the proposed negative feedback-loop regulation of MsrD that authors present in a section that is still not clearly presented, has been shown to exist for other genes conferring resistance to ribosome-targeting antibiotics. Lastly, the heavy editing has left unclear many sentences and paragraphs. Regardless of where the manuscript ends up being published, I encourage the authors to go through it and carefully polish it.

Few examples of specific unclear/problematic points:

- The summary lists findings of the work that only specialists may understand. In addition, it lacks a global message.
- Lane 68: "the exact molecular mechanism of action of ARE ABC-Fs remains unclear". In this new version, such mechanism is not addressed.
- Lanes 148-151: this paragraph seems unnecessary since mutant E. coli strains defective in efflux systems have been used for a long time to study mechanisms of antibiotic action.
- Supplementary fig. 1c: to really assess the expression level of the wt and mutant proteins, a normalizing control is needed (a Coomassie stained gel, for example?)
- Fig. 2a: would be easier to appreciate Ery-induced expression by plotting F/OD vs Ery concentration.
- Fig. 2c: would be clearer to show bar graphs representing F/OD +/- Ery, as shown in Supp fig. 2c
- Fig. 3: would be more logical to first show the toeprint in panel "c" and then the one shown in panel "a"
- Would make more sense to move Supp. fig. 3d as panel "e" of main Fig. 3. Current panel "e" can be a supplementary figure.
- Lanes 604-606: not clear what is the relevance of this comment. Identity of a single amino acid residue within a pentapeptide with otherwise different sequence does not corroborate a role in sensing the antibiotic structure.
- Experiments shown in Fig. 6a and supplementary fig. 6 would be much clearer by replotting the data as Fluorescence/OD vs. time at selected Ery concentrations. The fluorescence and OD data of the strains grown without antibiotic should be included.
- Lanes 837-839: unclear sentence
- Lanes 854-857: nothing novel or surprising about MsrD conferring resistance to non-inducing antibiotics. Just to mention an example, Ery-induced expression of ermC causes resistance to many antibiotics of the MLSb family among which many are not inducers.

Reviewer #3 (Remarks to the Author):

The authors have addressed all my comments.

Point-by-point responses to reviews of NCOMMS-22-30572A by Fostier *et al.* (reviewers 1- 3):

- **Reviewer #1 (Remarks to the Author):**

The authors have removed the section of the manuscript to which all my comments were addressed - I think this make the manuscript much more solid, but I hope the authors will continue to follow-up their splitting mechanism for ABCF, this would be very interesting for the community if it could be validated.

Authors' main response to Reviewer #1: We thank the Reviewer for his/her work to improve our manuscript and his/her wise comments. We will continue our investigation on the precise mechanism of action of the ARE ABC-F.

- **Reviewer #2 (Remarks to the Author):**

Comments on the revised manuscript

Main criticism:

Authors have made a great effort in improving the manuscript. Removing the section of the MsrD mode of action was a good decision. Including the in vitro toeprinting analysis of the selected MsrDL mutants further enhances the analysis of the nascent peptide and antibiotic mediated stalling which, in my opinion, remains the most solid and somehow novel portion of the work. However, the inducible expression of *msrD* by releasing transcriptional attenuation lacks novelty. Similarly, the proposed negative feedback-loop regulation of MsrD that authors present in a section that is still not clearly presented, has been shown to exist for other genes conferring resistance to ribosome-targeting antibiotics. Lastly, the heavy editing has left unclear many sentences and paragraphs. Regardless of where the manuscript ends up being published, I encourage the authors to go through it and carefully polish it.

Authors' main response to Reviewer #2: We thank the reviewer for this input and we appreciate that he/she thinks that we have improved the manuscript and that the analysis of the *msrD* induced ribosomal stalling in presence of antibiotic is novel and solid. Concerning the inducible expression of *msrD* by the attenuator, we agree that it uses a classical mechanism like many recent publications on leader peptide (van der Stel AX, Nat Commun. 2021 Sep 9;12(1):5340; Chadani Y, Mol Cell. 2017 Nov 2;68(3):528-539.e5) that show novelty in the stalling mechanism, but rarely on the attenuation mechanisms which are either Rho-independent or Rho-dependent when they are transcriptional attenuators.

We think the negative feedback-loop regulation of MsrD is presented accurately. We had stipulated that it was known for the ARE ABC-F Vga(A).

We have carefully reread our manuscript and invested substantial effort in making it clearer and more understandable.

Few examples of specific unclear/problematic points:

- The summary lists findings of the work that only specialists may understand. In addition, it lacks a global message.

We thank the reviewer for raising that point, but since we are on the second revision and the two other reviewers have accepted our revised version, we prefer to not make any major changes in the manuscript, including rewriting the abstract.

- Lane 68: “the exact molecular mechanism of action of ARE ABC-Fs remains unclear”. In this new version, such mechanism is not addressed.

Indeed, we removed some parts related to MsrD action on the ribosome. Nevertheless, because we are addressing MsrD regulation and have shown the action of some mutants of MsrD, it is appropriate to present the current knowledge of what is known about ARE ABC-F proteins, but according to the reviewer's suggestion we have rephrased the sentence in the revised manuscript.

- Lanes 148-151: this paragraph seems unnecessary since mutant E. coli strains defective in efflux systems have been used for a long time to study mechanisms of antibiotic action.

We respectfully disagree, this paragraph explains the bacteria model used for our study. We believe it is an important piece of information.

-Supplementary fig. 1c: to really assess the expression level of the wt and mutant proteins, a normalizing control is needed (a Coomassie stained gel, for example?)

We thank the reviewer for this important comment. All the protein extracts loaded on this gel were normalized to the OD 600 of the culture. We have added a Coomassie staining of the gel in the supplementary figure 1.

- Fig. 2a: would be easier to appreciate Ery-induced expression by plotting F/OD vs Ery concentration.

As explained in our previous response to the reviewers, we value this representation and since the other reviewers have accepted this representation, we prefer not to change it.

-Fig. 2c: would be clearer to show bar graphs representing F/OD +/- Ery, as shown in Supp fig. 2c

See previous comment and previous response to the reviewers.

- Fig. 3: would be more logical to first show the toeprint in panel “c” and then the one shown in panel “a”

Our logic is to first show the antibiotic specificity, then the specificity of the stalling. The other way around is also possible, but we would like to keep it this way, since it was validated by reviewers 1 and 3.

- Would make more sense to move Supp. fig. 3d as panel “e” of main Fig. 3. Current panel “e” can be a supplementary figure.

We thank the reviewer for this suggestion. We have moved panel “d” of the supplementary figure 3 to figure 3. We kept panel “e” in figure 3 to avoid too many changes, since reviewers 1 and 3 had accepted the previous version.

- Lanes 604-606: not clear what is the relevance of this comment. Identity of a single amino acid residue within a pentapeptide with otherwise different sequence does not corroborate a role in sensing the antibiotic structure.

These publications tested several peptides and showed that the leucine in position 3 is necessary for recognizing the bound antibiotic. We think that even if the rest of the sequence

is different, these observations are in accordance with what we proposed. We have rephrased the sentence to precise that this comment is only in accordance with our model.

- Experiments shown in Fig. 6a and supplementary fig. 6 would be much clearer by replotting the data as Fluorescence/OD vs. time at selected Ery concentrations. The fluorescence and OD data of the strains grown without antibiotic should be included.

In our previous revision, following the reviewer's request we have added two other concentrations of Ery for some of the mutants and we thank him/her for this important point. However, in this specific experiment we don't believe this new additional condition will provide any added value. The point of this figure is to show that plasmid expression of MsrD reduces the fluorescence level of the reporter. Figure 2 a and b already present the condition without ERY and clearly shows the level of basal fluorescence without ERY, in case this is a concern for the reviewer.

- Lanes 837-839: unclear sentence

Thanks to the reviewer for finding this error that arose during our editing. We have reformulated the sentence.

- Lanes 854-857: nothing novel or surprising about MsrD conferring resistance to non-inducing antibiotics. Just to mention an example, Ery-induced expression of ermC causes resistance to many antibiotics of the MLSb family among which many are not inducers.

We noted that is not surprising and accordingly, we didn't claim otherwise.

- **Reviewer #3 (Remarks to the Author):**

The authors have addressed all my comments.

Authors' main response to Reviewer #1: We thank the reviewer for his/her review of our manuscript and we think that he/she have help us to improve the quality and clarity of the manuscript.